# Tackling Decision Processes with Non-Cumulative Objectives using Reinforcement Learning

## Abstract

Markov decision processes (MDPs) are used to model a wide variety of applications ranging from game playing over robotics to finance. Their optimal policy typically maximizes the expected sum of rewards given at each step of the decision process. However, a large class of problems does not fit straightforwardly into this framework: Non-cumulative Markov decision processes (NCMDPs), where instead of the expected sum of rewards, the expected value of an arbitrary function of the rewards is maximized. Example functions include the maximum of the rewards or their mean divided by their standard deviation. In this work, we introduce a general mapping of NCMDPs to standard MDPs. This allows all techniques developed to find optimal policies for MDPs, such as reinforcement learning or dynamic programming, to be directly applied to the larger class of NCMDPs. Focusing on reinforcement learning, we show applications in a diverse set of tasks, including classical control, portfolio optimization in finance, and discrete optimization problems. Given our approach, we can improve both final performance and training time compared to relying on standard MDPs.

## 1 Introduction

Markov decision processes (MDPs) are used to model a wide range of applications where an agent iteratively interacts with an environment during a trajectory. Important examples include robotic control (Andrychowicz et al., 2020a), game playing (Mnih et al., 2015), or discovering algorithms (Mankowitz et al., 2023). At each time step $t$ of the MDP, the agent chooses an action based on the state of the environment and receives a reward $r_t$. The agent's goal is to follow an ideal policy that maximizes

$$\mathbb{E}_\pi \left[ \sum_{t=0}^{T-1} r_t \right], \tag{1}$$

where $T$ is the length of the trajectory and the expectation value is taken over both the agent's probabilistic policy $\pi$ and the probabilistic environment. There exist a multitude of established strategies for finding (approximately) ideal policies of MDPs, such as dynamic programming and reinforcement learning (Sutton & Barto, 2018).

A limitation of the framework of MDPs is the restriction to ideal policies that maximize Equation (1), while a large class of problems cannot straightforwardly be formulated this way. For example, in weakest-link problems, the goal is to maximize the minimum rather than the sum of rewards, e.g. in network routing one wants to maximize the minimum bandwidths along a path (Cui & Yu, 2023). In finance, the Sharpe ratio, i.e. the mean divided by the standard deviation of portfolio gains, is an important figure of merit of an investment strategy, since maximizing it will yield more risk-averse strategies than maximizing the sum of portfolio gains (Sharpe, 1966). We therefore require a method to tackle non-cumulative Markov decision processes (NCMDPs), where instead of the expected sum of rewards, the expectation value of an arbitrary function of the rewards is maximized. First steps in this direction have already been taken, but are currently limited to specific settings, e.g. by being restricted to only certain MDP solvers, deterministic environments, and restricted classes of non-cumulative objectives (for more details on these approaches see Section 4). The main contribution of this paper is twofold:

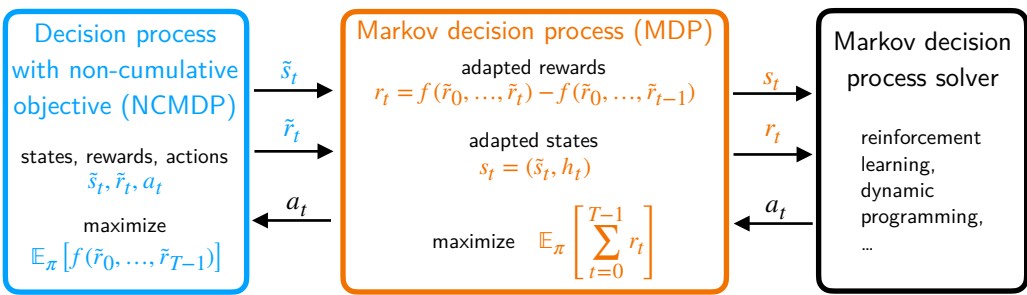

Figure 1: Mapping of a decision process with non-cumulative objective (blue) to a corresponding Markov decision process with adapted states and rewards (orange) that can be solved by standard methods (black). For details and notation, see Section 2.

- First, we provide a theoretical framework for a general and easy-to-implement mapping of NCMDPs to corresponding standard MDPs. This allows the direct application of advanced MDP solvers such as reinforcement learning or dynamic programming to tackle also NCMDPs (see Figure 1).
- Next, we perform numerical experiments focusing on reinforcement learning. We show applications in classical control problems, portfolio optimization, and discrete optimization problems using e.g. the non-cumulative $\max$ objective and the Sharpe ratio. Using our framework, we find improvements in both training time and final performance as compared to relying on standard MDPs.

## 2 THEORETICAL ANALYSIS

**Preliminaries**  In an MDP, an agent iteratively interacts with an environment during a trajectory. At each time step $t$, the agent receives the current state $s_t$ of the environment and selects an action $a_t$ by sampling from its policy $\pi(a_t|s_t)$, which is a probability distribution over all possible actions given a state. The next state of the environment $s_{t+1}$ and the agent's immediate reward $r_t$ are then sampled from the transition probability distribution $p(r_t, s_{t+1}|s_t, a_t)$, which depends on the MDP. This process repeats until it reaches a terminal state. The ideal policy of the agent maximizes the expected sum of immediate rewards, i.e. Equation (1). Note that we take this ideal policy to be part of the definition of an MDP to distinguish MDPs from NCMDPs with different ideal policies. For simplicity, we discuss only non-discounted, episodic MDPs. However, generalization to discounted settings is straightforward.

**Non-cumulative Markov decision processes**  In the following, we denote states and rewards related to NCMDPs by $\tilde{s}_t$ and $\tilde{r}_t$, respectively, to distinguish them from MDPs. We define NCMDPs equivalently to MDPs except for their ideal policies maximizing the expectation value of an arbitrary scalar function $f$ of the immediate rewards $\tilde{r}_t$ instead of their sum, i.e.

$$\mathbb{E}_\pi \left[ f(\tilde{r}_0, \tilde{r}_1, \ldots, \tilde{r}_{T-1}) \right]. \tag{2}$$

To accommodate trajectories of arbitrary finite length, we require $f$ to be a family of functions consisting of a function $f_t : \mathbb{R}^t \to \mathbb{R}$ for each $t \in \mathbb{N}$. For brevity, we denote $f_t(\tilde{r}_0, \tilde{r}_1, \ldots, \tilde{r}_{t-1}) = f(\tilde{r}_0, \tilde{r}_1, \ldots, \tilde{r}_{t-1})$. NCMDPs still have a Markovian transition probability distribution but their return, i.e. Equation (2), depends on the rewards in a non-cumulative and therefore non-Markovian way. Thus, NCMDPs are a generalization of MDPs. Due to this distinction, MDP solvers cannot straightforwardly be used for NCMDPs. To solve this problem, we describe a general mapping from an NCMDP $\tilde{M}$ to a corresponding MDP $M$ with adapted states and adapted rewards but the same actions. The mapping is chosen such that the optimal policy of $\tilde{M}$ is equivalent to the optimal policy of $M$. Therefore, a solution for $\tilde{M}$ can readily be obtained by solving $M$ with existing MDP solvers, as shown in Figure 1. In the following, we formalize these ideas.

**Definition 1.** Given an NCMDP $\tilde{M}$ with rewards $\tilde{r}_t$, states $\tilde{s}_t$, actions $a_t$, the Markovian state transition probability distribution $\tilde{p}(\tilde{r}_t, \tilde{s}_{t+1}|\tilde{s}_t, a_t)$, and non-cumulative objective function $f$, we

Table 1: Examples of non-cumulative objective functions $f$.

| $f(\tilde{r}_0, \ldots, \tilde{r}_t)$ | Additional state information $h_t$ | Adapted reward $r_t$ |
|---|---|---|
| $\max(\tilde{r}_0, \ldots, \tilde{r}_t)$ | $h_1 = \tilde{r}_0, h_{t+1} = \max(h_t, \tilde{r}_t)$ | $r_t = \max(0, \tilde{r}_t - h_t)$ |
| $\min(\tilde{r}_0, \ldots, \tilde{r}_t)$ | $h_1 = \tilde{r}_0, h_{t+1} = \min(h_t, \tilde{r}_t)$ | $r_t = \min(0, \tilde{r}_t - h_t)$ |
| Sharpe ratio $\frac{\mathrm{MEAN}(\tilde{r}_0,\ldots,\tilde{r}_t)}{\mathrm{STD}(\tilde{r}_0,\ldots,\tilde{r}_t)}$ | $h_0 = \begin{bmatrix} 0 \\ 0 \\ 0 \end{bmatrix}, h_{t+1} = \begin{bmatrix} \frac{h_t^{(2)}}{h_t^{(2)}+1}h_t^{(0)} + \frac{1}{h_t^{(2)}+1}\tilde{r}_t \\ \frac{h_t^{(2)}}{h_t^{(2)}+1}h_t^{(1)} + \frac{1}{h_t^{(2)}+1}\tilde{r}_t^2 \\ h_t^{(2)} + 1 \end{bmatrix}$ | $r_t = \frac{h_{t+1}^{(0)}}{\sqrt{h_{t+1}^{(1)} - h_{t+1}^{(0)2}}} - \frac{h_t^{(0)}}{\sqrt{h_t^{(1)} - h_t^{(0)2}}}$ |
| $\max\limits_{k \in [-1,t]} \sum_{i=0}^{k} \tilde{r}_i$ | $h_0 = 0, h_{t+1} = \max(0, h_t - \tilde{r}_t)$ | $r_t = \max(0, \tilde{r}_t - h_t)$ |
| $\tilde{r}_0 \tilde{r}_1 \ldots \tilde{r}_t$ | $h_0 = 1, h_{t+1} = \tilde{r}_t h_t$ | $r_t = h_{t+1} - h_t$ |
| Harmonic mean $\frac{1}{\frac{1}{\tilde{r}_0} + \cdots + \frac{1}{\tilde{r}_t}}$ | $h_0 = 0, h_{t+1} = h_t + \frac{1}{\tilde{r}_t}$ | $r_t = \frac{1}{h_{t+1}} - \frac{1}{h_t}$ |
| $\delta^t \sum_{k=0}^{t} \tilde{r}_k, \\ \delta \in (0,1)$ | $h_0 = \begin{bmatrix} 0 \\ 0 \end{bmatrix}, h_{t+1} = \begin{bmatrix} h_t^{(0)} + \tilde{r}_t \\ h_t^{(1)} + 1 \end{bmatrix}$ | $r_t = \delta^{h_{t+1}^{(1)}} h_{t+1}^{(0)} - \delta^{h_t^{(1)}} h_t^{(0)}$ |
| $\frac{1}{t+1} \sum_{k=0}^{t} \tilde{r}_k$ | $h_0 = \begin{bmatrix} 0 \\ 0 \end{bmatrix}, h_{t+1} = \begin{bmatrix} h_t^{(0)} + \tilde{r}_t \\ h_t^{(1)} + 1 \end{bmatrix}$ | $r_t = \frac{1}{h_{t+1}^{(1)}+1} h_{t+1}^{(0)} - \frac{1}{h_t^{(1)}+1} h_t^{(0)}$ |

define a *corresponding* MDP $M$ with rewards $r_t$, states $s_t$, the same actions $a_t$, and state transition probability distribution $p(r_t, s_{t+1}|s_t, a_t)$ with functions $\rho$ and $u$, and vectors $h_t$ such that

$$r_t = f(\tilde{r}_0, \ldots, \tilde{r}_t) - f(\tilde{r}_0, \ldots, \tilde{r}_{t-1}) = \rho(h_t, \tilde{r}_t), \tag{3}$$

$$s_t = (\tilde{s}_t, h_t) \text{ with } h_{t+1} = u(h_t, \tilde{r}_t), \tag{4}$$

$$p\big(r_t, s_{t+1}=(\tilde{s}_{t+1}, h_{t+1})|s_t=(\tilde{s}_t, h_t), a_t\big) = \sum_{\tilde{r}_t} \tilde{p}(\tilde{r}_t, \tilde{s}_{t+1}|\tilde{s}_t, a_t)\delta_{h_{t+1}, u(h_t, \tilde{r}_t)}\delta_{r_t, \rho(h_t, \tilde{r}_t)}, \tag{5}$$

where $\delta$ is the Kronecker delta. For continuous probability distributions, the sum should be replaced by an integral and the $\delta$ by Dirac delta functions.

We now provide some intuition for this definition: Equation (3) ensures that the return of $M$, i.e. $\sum_{t=0}^{T-1} r_t$, is equal to the return of $\tilde{M}$, i.e. $f(\tilde{r}_0, \ldots, \tilde{r}_{T-1})$. To compute the immediate rewards $r_t$ of $M$ in a purely Markovian manner, we need access to information about the previous rewards of the trajectory. This can be achieved by extending the state space with $h_t$, which preserves all necessary information about the reward history. The function $u$ in Equation (4) updates this information at each time step. Finally, the Kronecker deltas in Equation (5) 'pick' all the possible $\tilde{r}_t$ that result in the same $h_{t+1}$ and $r_t$. For example, for $f(\tilde{r}_0, \ldots, \tilde{r}_t) = \min(\tilde{r}_0, \ldots, \tilde{r}_t)$, we can choose $r_0 = h_1 = \tilde{r}_0$ followed by $h_t = u(h_t, \tilde{r}_t) = \min(h_t, \tilde{r}_t)$, and $r_t = \rho(h_t, \tilde{r}_t) = \min(0, \tilde{r}_t - h_t)$. More examples are shown in Table 1.

Due to Definition 1, there is a mapping between trajectories of NCMDPs and corresponding MDPs, which we will exploit to use the same policy for both decision processes:

**Definition 2.** A trajectory $\tilde{\mathcal{T}} = (\tilde{s}_0, a_0, \tilde{r}_0, \ldots)$ of an NCMDP $\tilde{M}$ is *mapped* onto a trajectory $\mathcal{T} = \mathrm{map}\big(\tilde{\mathcal{T}}\big) = (s_0, a_0, r_0, \ldots)$ of a corresponding MDP $M$ by calculating $r_t$ according to Equation (3), setting $s_t = (\tilde{s}_t, h_t)$ with $h_t$ calculated according to Equation (4), and keeping the actions $a_t$ the same.

Note that multiple $\tilde{\mathcal{T}}$ potentially map to the same $\mathcal{T}$ since $\rho$ and $u$, and, therefore, the map operation are not necessarily injective. We can now state the main result of this manuscript:

**Theorem 1.** *Consider an NCMDP $\tilde{M}$ and a corresponding MDP $M$, both defined as in Definition 1. Then, given a policy $\pi(a_t|s_t)$ of $M$, the expected return of $M$ is equal to the expected return of $\tilde{M}$, i.e.*

$$\mathbb{E}_\pi \left[ \sum_{t=0}^{T-1} r_t \right] = \mathbb{E}_\pi \left[ f(\tilde{r}_0, \ldots, \tilde{r}_{T-1}) \right], \tag{6}$$

*if $\pi(a_t|s_t)$ is used to interact with $\tilde{M}$ as follows: At each time step $t$, map the current trajectory $\tilde{\mathcal{T}}$ of $\tilde{M}$ to a trajectory $\mathcal{T} = \mathrm{map}\left(\tilde{\mathcal{T}}\right)$ of $M$ as described in Definition 2, thereby finding $s_t$. Then choose an action according to $\pi(a_t|s_t)$.*

For a proof of this theorem see Appendix A. Since $M$ is a standard MDP, we can find its optimal policy using standard methods and are guaranteed that it will also be the ideal policy of $\tilde{M}$. More generally, also non-ideal policies will yield the same return for both decision processes. By mapping the NCMDP to an MDP, all guarantees available for MDP solvers, e.g. convergence proofs for Q-learning (Watkins & Dayan, 1992), or the policy gradient theorem for policy-based methods (Sutton et al., 1999), directly apply also to NCMDPs.

We demonstrate the need for the additional state information $h_t$ in the two-step NCMDP depicted in Figure 2, which captures the main conceptual difficulties in solving NCMDPs: A non-cumulative objective in a stochastic environment. Here, the goal is to maximize the expected minimum of the rewards, i.e. $\min(\tilde{r}_0, \tilde{r}_1)$. If $\tilde{r}_0 = 1$ in the first step, the ideal policy is to choose $a_1$ in the second step, while if $\tilde{r}_0 = -1$ the agent should choose $a_0$. Therefore, the choice of the ideal action depends on information contained in the past rewards, which is captured by $h_1 = \tilde{r}_0$. We show the MDP constructed from the NCMDP as described above in the Appendix in Figure A1 and its state-action value function found by value iteration in Table A1. Our method finds the optimal policy, while previous methods for solving NCMDPs fail even in this simple example because they neglect the extra state information (see Table A2).

The question remains how to find the functions $u$ and $\rho$ given a new objective $f$. In principle, methods developed based on past (Bacchus et al., 1996; 1997) or future (Thiébaux et al., 2006) linear temporal logic could be used. However, these require an expensive computation over all possible states of the MDP making them inefficient for online learning. There is always at least one MDP corresponding to a given NCMDP since we can take $h_t = [\tilde{r}_0, \ldots, \tilde{r}_{t-1}]$ with $u(h_t, \tilde{r}_t) = [h_t, \tilde{r}_t]$ and $\rho(h_t, \tilde{r}_t) = f([h_t, \tilde{r}_t]) - f(h_t)$. However, this is not ideal because it leads to a state size that grows linearly with the trajectory length. In Appendix B.1, we provide a necessary and sufficient condition for objectives $f$ with constant size additional state information $h_t$. However, this condition might be difficult to check in practice. Therefore, we also provide a sufficient, but not necessary, condition that is easier to verify and covers a large class of functions, e.g. all functions in Table 1. For functions in this class, we also provide an explicit construction of $u$, $\rho$, and $h_t$ (see Appendix B.2). In practice, we have empirically observed that a simple analytical consideration leads to $h_t$ of a small and constant size for each of the objectives $f$ considered in this manuscript, which is desirable for efficient learning and integration with standard function estimators such as neural networks.

## 3 EXPERIMENTS

From an implementation perspective, our scheme of mapping NCMDPs to MDPs requires minimal effort, since we can treat both the NCMDP and the used MDP solver as black boxes by simply adding a layer between them, as shown in Figure 1. In addition, our treatment facilitates online learning and does not require any computationally expensive preprocessing of the NCMDP. This opens the door for researchers with specific domain knowledge, who are not necessarily experts in reinforcement learning, to use standard libraries such as `stable-baselines3` (Raffin et al., 2021) to solve their non-cumulative problems. Conversely, it allows reinforcement learning experts to quickly tackle existing environments using our method.

There are three kinds of experiments presented in this manuscript:

1. We show that our method can find the optimal policy in a proof-of-principle stochastic NCMDPs where previous methods fail (see previous section, Figure 2).

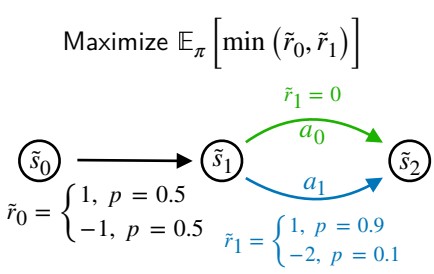

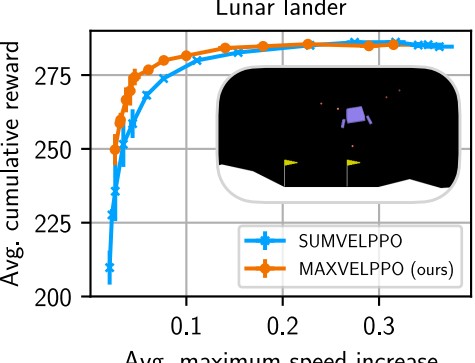

Figure 2: Two-step decision process with non-cumulative objective. In the first step, the agent has only one action available and receives a probabilistic reward. In the second step, the agent can choose between two actions.

Figure 3: Trade-off between average cumulative reward in the Lunar lander environment (inset) and maximum speed increase for the non-cumulative MAXVELPPO and the cumulative SUMVELPPO algorithm.

2. In Appendix C, we compare our method with the only previous approach for solving NCMDPs with objectives other than the max function, which was introduced by Cui & Yu (2023), and is only applicable with more restrictive objectives $f$, in deterministic environments, and in conjunction with Q-learning methods. We find in our experiments that even in this restricted setting, our more general method performs slightly better.

3. In the following sections, we show applications in real-world tasks where previous methods used for solving special cases of NCMDPs cannot be applied. In lack of a general method for solving NCMDPs, state-of-the-art approaches to these real-world problems have so far relied on approximate solutions based on standard MDPs which we use as a baseline to compare our method to.

We provide details on hyperparameters, compute resources, and training curves for all experiments in Appendix D.

### 3.1 CLASSICAL CONTROL

As a first use case of our method, we train a reinforcement learning agent in the Lunar lander environment of the gymnasium (Towers et al., 2023) library. The agent controls a spacecraft with four discrete actions corresponding to different engines while being pushed by a stochastic wind. Immediate positive rewards $r_t$ are given for landing the spacecraft safely with small negative rewards given for using the engines. A realistic goal when landing a spacecraft is to not let the spacecraft get too fast, e.g. to avoid excessive frictional heating. Therefore, we define an NCMDP where the agent is penalized for its maximum speed during a trajectory, i.e. we try to maximize

$$\mathbb{E}_\pi \left[ \sum_{t=0}^{T-1} r_t - c \max\left(v_0, \ldots, v_{T-1}\right) \right], \tag{7}$$

where $v_t$ is the speed of the agent at time $t$ and $c$ defines a trade-off between minimizing the maximum speed and the other goals of the agent. We train RL agents using Proximal Policy Optimization (PPO) (Schulman et al., 2017) on the MDP constructed from the NCMDP as described above (MAXVELPPO). To show the trade-off between the sum of the agents' original cumulative rewards $\sum_{t=0}^{T-1} r_t$ and their maximum speed increase during a trajectory, we train agents for different $c$. Each value of $c$ corresponds to a marker in Figure 3. We compare our results to an RL agent with a similar but cumulative objective maximizing $\mathbb{E}_\pi \left[ \sum_{t=0}^{T-1} (r_t - c v_t^2) \right]$ (SUMVELPPO). As shown in Figure 3, the non-cumulative MAXVELPPO agent is consistently able to find a better trade-off than the cumulative SUMVELPPO agent. All experiments were performed with 5 agents using different seeds. We plot average results and standard deviation as error bars for both the cumulative reward and the speed increase (mostly hidden behind markers).

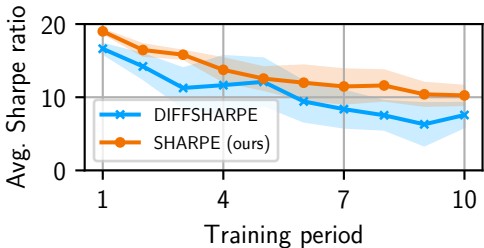

| | DIFFSHARPE | SHARPE |
|---|---|---|
| Training | $10.51 \pm 1.65$ | $13.33 \pm 1.14$ |
| Evaluation | $1.43 \pm 0.18$ | $1.35 \pm 0.15$ |
| Test | $0.53 \pm 0.26$ | $0.51 \pm 0.09$ |

Table 2: Average Sharpe ratios.

Figure 4: Portfolio optimization. Sharpe ratio during 10 different training periods for algorithms maximizing the cumulative differential Sharpe ratio (DIFFSHARPE) and the non-cumulative exact Sharpe ratio (SHARPE).

Our method presented here could be applied to similar use cases in other classical control problems, such as teaching a robot to reach a goal while minimizing the maximum impact forces on its joints or the forces its motors need to apply.

### 3.2 PORTFOLIO OPTIMIZATION WITH SHARPE RATIO AS OBJECTIVE

Next, we consider the task of portfolio optimization where an agent decides how to best invest its assets across different possibilities. A common measure for a successful investment strategy is its Sharpe ratio (Sharpe, 1966)

$$\frac{\text{MEAN}(\tilde{r}_0, \ldots, \tilde{r}_{T-1})}{\text{STD}(\tilde{r}_0, \ldots, \tilde{r}_{T-1})}, \tag{8}$$

where $\tilde{r}_t = (P_{t+1} - P_t)/P_t$ are the simple returns and $P_t$ is the portfolio value at time $t$. By dividing through the standard deviation of the simple returns, the agent is discouraged from risky strategies with high volatility. As the Sharpe ratio is non-cumulative, reinforcement learning strategies so far needed to fall back on the approximate differential Sharpe ratio as a reward (Moody et al., 1998; Moody & Saffell, 2001). However, using the methods developed in this paper we can directly maximize the exact Sharpe ratio. We perform experiments on an environment as described by Sood et al. (2023) where an agent trades on the 11 different S&P500 sector indices between 2006 and 2021. The states contain a history of returns of each index and different volatility measures. Actions are the continuous relative portfolio allocations for each day. The experiment described by Sood et al. (2023) consists of training 5 agents with different seeds over a 5-year period, periodically evaluating their performance on the following year, and testing the best-performing agent in the year after that. Then, the time period is shifted by one year into the future resulting in a total of 10 time periods. We re-implement this experiment by training agents using PPO with the cumulative differential Sharpe ratio (DIFFSHARPE) or the exact Sharpe ratio (SHARPE) as their objective. As depicted in Figure 4, the SHARPE algorithm significantly outperforms the DIFFSHARPE algorithm in the training years. However, as shown in Table 2, on the evaluation and test years there is no significant difference indicating over-fitting of the agents' policies to the years they are trained on. Nonetheless, we expect training on the exact SHARPE ratio to give consistently better results if the problem of over-fitting is solved, e.g. by using realistic stock-market simulators which are for example being developed using Generative Adversarial Networks (Li et al., 2020). The experiments were performed with 5 sets of 5 seeds each and the standard deviation of different sets of seeds is reported.

While the Sharpe ratio is most widely adopted in finance, our method opens up the possibility to maximize it also in other scenarios where risk-adjusted rewards are desirable, i.e. all problems where consistent rewards with low variance are more important than a higher cumulative reward. For example, in chronic disease management, maintaining stable health metrics is preferable to sporadic improvements. In emergency or customer service, ensuring predictable response times is often more important than occasional fast responses mixed with slow ones.

## 3.3 DISCRETE OPTIMIZATION PROBLEMS

Next, we consider a large class of applications where RL is commonly used: Problems where the agent iteratively transforms a state by its actions to find a state with a lower associated cost. These problems are common in scientific applications such as physics, e.g. to reduce the length of quantum logic circuits (Fösel et al., 2021), or chemistry, e.g. for molecular discovery (Zhou et al., 2019). Another prominent example is the discovery of new algorithms (Mankowitz et al., 2023). Intuitively, these problems can be understood as searching for the state with the lowest cost within an equivalence class defined by all states that can be reached from the start state by the agent's actions.

Concretely, we consider the class of discrete optimization problems equipped with a scalar cost function $c(\tilde{s}_t)$ and the immediate rewards $\tilde{r}_t = c(\tilde{s}_t) - c(\tilde{s}_{t+1})$. Additionally, we are interested in the state with the lowest cost found during a trajectory, i.e. the goal is to maximize

$$\mathbb{E}_\pi \left[ c(\tilde{s}_0) - \min_{k \in [0, T-1]} c(\tilde{s}_k) \right] = \mathbb{E}_\pi \left[ \max_{k \in [-1, T-1]} \sum_{t=0}^{k} \tilde{r}_t \right]. \tag{9}$$

We conjecture that maximizing Equation (9) will yield better results than maximizing $\mathbb{E}_\pi \left[ \sum_{t=0}^{T-1} \tilde{r}_t \right]$ due to the following reasons:

1. The agent does not need to learn an optimal stopping point.

2. Considering only the rewards up to the minimum found cost might decrease the variance of the gradient estimate.

3. The agent does not receive negative rewards for trying to escape a local cost minimum during a trajectory and is therefore not discouraged from exploring. This leads to learning difficult optimization strategies requiring an intermittent cost increase more easily.

**Peak environment** To facilitate an in-depth analysis, we first consider a toy environment with the cost function depicted in the inset of Figure 5 (a). The cost function was chosen to be simple while still requiring intermittently cost-increasing actions of the optimal policy. Each trajectory lasts 10 steps and the agent's actions are stepping to the left, right, or doing nothing. To minimize the number of hyperparameters, we use the REINFORCE algorithm (Williams, 1992) with a tabular policy. We compare agents trained with the non-cumulative objective Equation (9) (MAXREINFORCE) with agents that maximize the cumulative rewards (REINFORCE). As shown in Figure 5 (a), the MAXREINFORCE agent trains much faster. Two possible sources of this speed-up are a different, more advantageous direction of the gradient updates and a reduced variance when estimating these gradients. To investigate which is the case, we periodically stop training and run $n = 1000$ trajectories with a fixed policy. We then compute the empirical variance of the gradient update inspired by Kaledin et al. (2022) as

$$\text{VAR} = \frac{1}{n} \sum_{i=0}^{n-1} \|\vec{g}_i\|_2^2 - \left\| \frac{1}{n} \sum_{i=0}^{n-1} \vec{g}_i \right\|_2^2, \tag{10}$$

where $\vec{g}_i$ is the gradient from a single trajectory derived either by the MAXREINFORCE or the REINFORCE algorithm. As this variance scales with the squared magnitude of the average gradients, we normalize it by this value to ensure a fair comparison between the two algorithms. In Figure 5 (b), we show that in the initial phase of training, the MAXREINFORCE algorithm significantly reduces variance. To compare the direction of the gradients we use the same data to compute the normalized average gradients of both algorithms and show their dot product in Figure 5 (c). We find that the gradients are correlated (i.e. the dot product is bigger than zero) but not the same. Therefore, we conclude that the training speed-up is derived both from lower variance and a better true gradient direction. All results reported are averaged over 10 seeds with standard deviations plotted shaded.

**ZX-diagrams** As a real-world use case, we consider the simplification of ZX-diagrams, which are graph representations of quantum processes (Coecke & Kissinger, 2017) with applications e.g. in the compilation of quantum programs (Duncan et al., 2020; Riu et al., 2023). An example of a typical ZX-diagram is shown in the inset of Figure 6 (b). We consider the environment described by Nägele & Marquardt (2024), where the cost function of a diagram is given by its node number, the start states

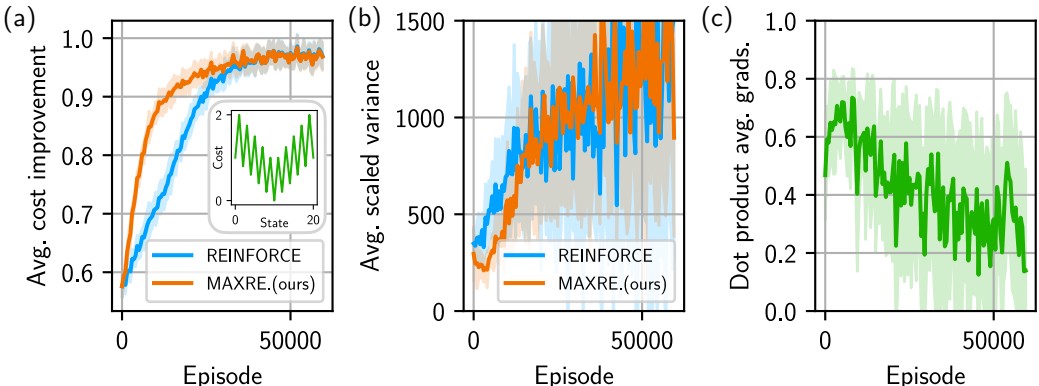

Figure 5: Peak environment. (a) Cost improvement during the trajectory against training episodes for the cumulative REINFORCE agent and the MAXREINFORCE agent maximizing Equation (9). The inset shows the cost function of the environment. (b) Empirical variance of both agents against training progress. (c) Estimated dot product of the average gradient of both agents against training progress.

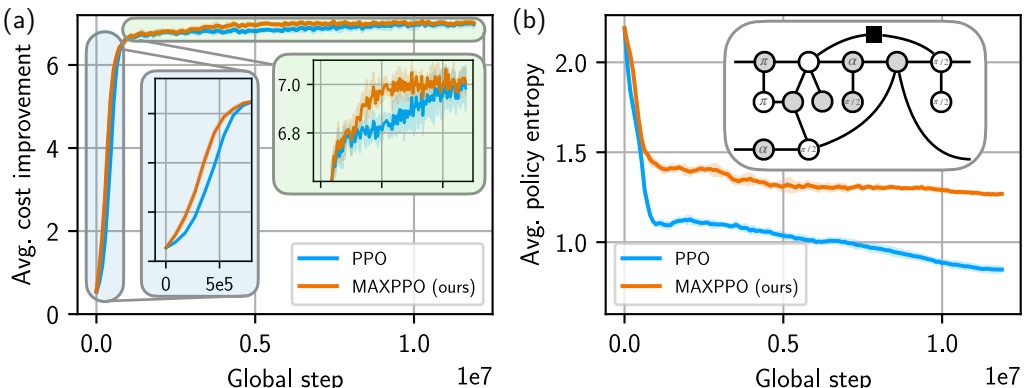

Figure 6: ZX environment. (a) Cost improvement during the trajectory against total steps taken in the environment for the cumulative REINFORCE agent and the MAXREINFORCE agent maximizing Equation (9). Left inset: Zoom-in to the initial training phase. Right inset: Zoom-in to the final training phase. (b) Entropy of the agents' policies against total steps taken in the environment. Inset: A typical ZX-diagram.

are randomly sampled ZX-diagrams, and the actions are a set of local graph transformations. In total, there are 6 actions per node and 6 actions per edge in the diagram. This is a challenging reinforcement learning task that requires the use of graph neural networks to accommodate the changing size of the state and action space. We use PPO to train agents to maximize either Equation (9) (MAXPPO) or the cumulative reward (PPO) with a trajectory length of 20 steps. As shown in Figure 6 (a) the MAXPPO agent initially trains faster than the PPO agent (left inset). This is likely due to the reduced variance and different gradient direction as described above. The PPO agent then shortly catches up, but ultimately requires about twice as many training steps to reach optimal performance as the MAXPPO agent (right inset). We argue that this is because the MAXPPO agent is better at exploring and therefore learning difficult optimization strategies (reason 3 above). This is captured by the entropy of the MAXPPO agent's policy staying much higher than the entropy of the PPO agent, as shown in Figure 6 (b). All reported results are averaged over 5 seeds with standard deviations plotted shaded.

**Quantum error correction** Next, we focus on optimization problems where the starting state is always the same and the optimal stopping point is known. The specific tasks we consider are the search for quantum programs to either find new quantum error correction codes or to prepare logical

Table 3: Results for quantum error correction environments.

| Task | Area under training curve $\frac{\text{MAXPPO}}{\text{PPO}}$ | |
|---|---|---|
| | Best of 10 | Mean of 10 |
| $[[5,1,3]]_e$ | **1.0004** | **1.0001 ± 0.0008** |
| $[[7,1,3]]_e$ | 0.9993 | 0.9994 ± 0.0003 |
| $[[9,1,3]]_e$ | 0.9994 | 0.9994 ± 0.0003 |
| $[[10,2,4]]_e$ | **1.0013** | **1.0023 ± 0.0017** |
| $[[12,2,4]]_e$ | 0.9993 | 0.9991 ± 0.0007 |
| $[[5,1,3]]_s$ | **1.0419** | **1.0666 ± 0.0407** |
| $[[7,1,3]]_s$ | **1.1824** | **1.2090 ± 0.0623** |
| $[[9,1,3]]_s$ | **1.0060** | **1.0905 ± 0.1079** |
| $[[15,1,3]]_s$ | **1.1382** | **1.0309 ± 0.0806** |
| $[[17,1,5]]_s$ | **1.4748** | **1.6218 ± 0.2144** |

states of a given quantum error correction code - both critical for the eventual realization of quantum computation (Terhal, 2015). In both cases, the agent iteratively adds elementary quantum logic gates until the program delivers the desired result. The agents' policies are encoded by standard multilayer perceptrons. We train agents on five different sized problems of both tasks. For details on the used environment, see Zen et al. (2024) and Olle et al. (2023). To obtain a single performance measure encompassing both final performance and training speed, we continuously evaluate the mean cost improvement of the agents and average it over the training process as suggested by Andrychowicz et al. (2020b). Intuitively, this measure can be understood as the area under the agent's training curve. In Table 3, we report the quotient of this performance measure of the MAXPPO algorithm and the PPO algorithm for both the best of 10 and the mean of 10 trained agents. The different tasks are denoted by three integers, as customary in the quantum error correction community. We find that the MAXPPO algorithm performs similarly to PPO in quantum error correction code discovery (subscript e), and significantly better in logical state preparation (subscript s). We argue that the large degeneracy present in the solution space of the code discovery task diverts the more exploratory MAXPPO, lowering its performance to the level of PPO.

**Limitation** When applying MAXPPO to discrete optimization problems with long trajectories in the many hundreds of steps, we empirically observed an initially slow learning speed. This could be due to the agent initially mostly increasing cost and therefore receiving zero reward for almost the entire trajectory. A possible solution could be to dynamically adjust the trajectory length in these problems going from shorter to longer trajectories during the training process.

## 4 RELATED WORK

A special case of NCMDPs is first considered by Quah & Quek (2006), who adapt Q-learning to the max objective by redefining the temporal difference error of their learning algorithms and demonstrate their algorithms on an optimal stopping problem. However, they do not adapt state space and do not provide theoretical convergence guarantees. Gottipati et al. (2020) rediscover the same algorithm, apply it to molecule generation, and provide convergence guarantees for their method, while Eyckerman et al. (2022) consider the same algorithm for application placement in fog environments. Cui & Yu (2023) finally show a shortcoming of the method used in the above papers: It is guaranteed to converge to the optimal policy only in deterministic settings. Additionally, they provide convergence guarantees of Q-learning in deterministic environments for a larger class of functions $f$, focusing on the min function for network routing applications. In Appendix B.1 we show that the class of objectives $f$ considered by Cui & Yu (2023) is a subclass of all objectives that lead to constant size extra state information $h_t$ using our method. Independently, the max function is also used in the field of safety reinforcement learning in deterministic environments both for Q-learning (Fisac et al., 2015; 2019; Hsu et al., 2021) and policy-based reinforcement learning (Yu et al., 2022). Moflic & Paler (2023) use a reward function with parameters that depend on the past rewards of the trajectory to tackle quantum circuit optimization problems that require large intermittent negative

rewards, albeit without providing theoretical convergence guarantees. Additionally, Wang et al. (2020) investigate planning problems with non-cumulative objectives within deterministic settings. They provide provably efficient planning algorithms for a large class of functions $f$ by discretizing rewards and appending them to the states. Recently, Veviurko et al. (2024) showed how to use the $\max$ objective also in probabilistic settings by augmenting state space and provide convergence guarantees for both Q-learning and policy-based methods. They then show experiments with their algorithm yielding improvements in MDPs with shaped rewards. However, they do not redefine the rewards, which requires adaptation of the implementation of their MDP solvers. Our method described above reduces to an effectively equivalent algorithm to theirs in the special case of the $\max$ objective.

A limitation of all works discussed above is that they require a potentially complicated adaptation of their reinforcement learning algorithms and only consider specific MDP solvers or specific non-cumulative objectives. They are also limited to deterministic settings, except for Veviurko et al. (2024), which consider only the $\max$ objective.

## 5 DISCUSSION & CONCLUSION

In this work, we described a mapping from a decision process with a general non-cumulative objective (NCMDP) to a standard Markov decision process (MDP) applicable in deterministic and probabilistic settings. As our method is agnostic to the algorithm used to solve the resulting MDP, it works for arbitrary action spaces and in conjunction with both off- and on-policy algorithms. Its implementation is straightforward and directly enables solving NCMDPs with state-of-the-art MDP solvers, allowing us to show improvements in a diverse set of tasks such as classical control problems, portfolio optimization, and discrete optimization problems. Note that these improvements are achieved without adding a single additional hyperparameter to the solving algorithms.

In further theoretical work, a full constructive classification of objective functions $f$ with constant-size extra state information $h_t$ would be desirable. From an applications perspective, there are a lot of interesting objectives with non-cumulative $f$ that could not be maximized so far. For example, the geometric mean could be used to maximize average growth rates, or the function $f(\tilde{r}_0, \ldots, \tilde{r}_t) = \delta^t \sum_{k=0}^{t} \tilde{r}_k, \delta \in (0, 1)$ could be used to define an exponential trade-off between trajectory length and cumulative reward in settings where long trajectories are undesirable. We believe that a multitude of other applications with non-cumulative objectives are still unknown to the reinforcement learning (RL) community, and conversely, that researchers working on non-cumulative problems are not aware of RL, simply because these two concepts could not straightforwardly be unified so far. This manuscript offers the exciting possibility of discovering and addressing this class of problems still unexplored by RL.

## REPRODUCIBILITY

Anonymized code including exact hyperparameters and random seeds, generated data, trained agent weights, and instructions for running the code for all presented experiments are available for download at `https://osf.io/ajwmk/?view_only=38b116ac6633496a83657aeff43db34a`. Upon acceptance, we will make the code and data publicly available. Furthermore, we state or provide references to all used hyperparameters and computation times for all experiments in Appendix D.

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

# Appendices

## A  PROOF OF THEOREM 1

First, we find that the probability $p(\mathcal{T})$ of a trajectory $\mathcal{T}$ occurring in $M$ is the sum of the probabilities $p(\tilde{\mathcal{T}})$ of all possible trajectories $\tilde{\mathcal{T}}$ occurring in $\tilde{M}$ that map to $\mathcal{T}$ if $\pi$ is used as a policy for $\tilde{M}$ as described above:

$$
\begin{aligned}
p(\mathcal{T}) &= p(s_0) \prod_{t=0}^{T-1} \pi(a_t|s_t) p(r_t, s_{t+1}|s_t, a_t) \\
&= p(\tilde{s}_0) \prod_{t=0}^{T-1} \pi(a_t|s_t) \sum_{\tilde{r}_t} \tilde{p}(\tilde{r}_t, \tilde{s}_{t+1}|\tilde{s}_t, a_t) \delta_{h_{t+1}, u(h_t, \tilde{r}_t)} \delta_{r_t, \rho(h_t, \tilde{r}_t)} \\
&= \sum_{\mathrm{map}(\tilde{\mathcal{T}})=\mathcal{T}} p(\tilde{s}_0) \prod_{t=0}^{T-1} \pi(a_t|s_t) \tilde{p}(\tilde{r}_t, \tilde{s}_{t+1}|\tilde{s}_t, a_t) = \sum_{\mathrm{map}(\tilde{\mathcal{T}})=\mathcal{T}} p(\tilde{T}),
\end{aligned}
\tag{11}
$$

where $\sum_{\mathrm{map}(\tilde{\mathcal{T}})=\mathcal{T}}$ is the sum over all trajectories of $\tilde{M}$ that map to $\mathcal{T}$. The second to last step can be shown through induction. Therefore, we find

$$
\begin{aligned}
\mathbb{E}_\pi \left[ \sum_{t=0}^{T-1} r_t \right] &= \sum_{\mathcal{T}} p(\mathcal{T}) \sum_{t=0}^{T-1} r_t = \sum_{\mathcal{T}} \sum_{\mathrm{map}(\tilde{\mathcal{T}})=\mathcal{T}} p(\tilde{\mathcal{T}}) f(\tilde{r}_0, \dots, \tilde{r}_{T-1}) \\
&= \sum_{\tilde{\mathcal{T}}} p(\tilde{\mathcal{T}}) f(\tilde{r}_0, \dots, \tilde{r}_{T-1}) = \mathbb{E}_\pi \left[ f(\tilde{r}_0, \dots, \tilde{r}_{T-1}) \right],
\end{aligned}
\tag{12}
$$

where in the second step we used Equations (3) and (11), and in the third step that each $\tilde{\mathcal{T}}$ maps to a unique $\mathcal{T}$. $\qquad\square$

## B  OBJECTIVES $f$ WITH CONSTANT SIZE EXTRA STATE INFORMATION $h_t$

### B.1  NECESSARY AND SUFFICIENT CONDITION FOR CONSTANT SIZE EXTRA STATE INFORMATION $h_t$

First, note that if we can predict $f(\tilde{r}_0, \dots, \tilde{r}_t)$ in a Markovian manner, we can also predict $f(\tilde{r}_0, \dots, \tilde{r}_t) - f(\tilde{r}_0, \dots, \tilde{r}_{t-1})$ in a Markovian manner by adding just one more dimension to our state $h_t$, i.e. $f(\tilde{r}_0, \dots, \tilde{r}_{t-1})$. Therefore, we focus on predicting $f(\tilde{r}_0, \dots, \tilde{r}_t)$ in the following. Rewriting Definition 1, we find using

$$
f(\tilde{r}_0, \dots, \tilde{r}_t) - f(\tilde{r}_0, \dots, \tilde{r}_{t-1}) = \rho\left(\tilde{r}_t, u\left(\tilde{r}_{t-1}, u\left(\tilde{r}_{t-2}, \dots\right)\right)\right)
\tag{13}
$$

or equivalently

$$
f(\tilde{r}_0, \dots, \tilde{r}_t) = \rho'\left(\tilde{r}_t, u'\left(\tilde{r}_{t-1}, u'\left(\tilde{r}_{t-2}, \dots\right)\right)\right),
\tag{14}
$$

with

$$
u'(\tilde{r}_t, h'_t) = u'\left(\tilde{r}_t, (h_t, f_{t-1})\right) = \left[u(\tilde{r}_t, h_t), f_{t-1} + \rho(\tilde{r}_t, h_t)\right],
\tag{15}
$$

and

$$
\rho'(\tilde{r}_t, h'_t) = \rho(\tilde{r}_t, h_t) + f_{t-1}.
\tag{16}
$$

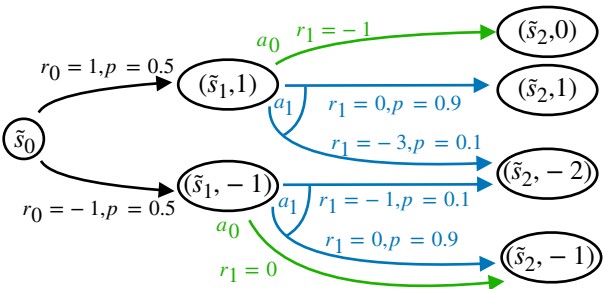

Figure A1: Markov decision process constructed from the non-cumulative Markov decision process depicted in Figure 2.

Table A1: Q-function $Q$ of the MDP depicted in Figure A1 (which corresponds to the NCMDP of Figure 2) found by value iteration using our method. The resulting policy is optimal and results in an expected return of $-0.15$.

| State $s$ | Action $a$ | $Q(s,a)$ |
|---|---|---|
| $\tilde{s}_0$ | - | $-0.15$ |
| $(\tilde{s}_1, 1)$ | $a_0$ | $-1$ |
| $(\tilde{s}_1, 1)$ | $a_1$ | $-0.3$ |
| $(\tilde{s}_1, -1)$ | $a_0$ | $0$ |
| $(\tilde{s}_1, -1)$ | $a_1$ | $-0.1$ |

Table A2: Q-function $Q'$ of the MDP depicted in Figure 2 found by value iteration using the method of Cui & Yu (2023). The resulting policy is not optimal and results in an expected return of $-0.5$.

| State $\tilde{s}$ | Action $a$ | $Q'(\tilde{s},a)$ |
|---|---|---|
| $\tilde{s}_0$ | - | $-0.5$ |
| $\tilde{s}_1$ | $a_0$ | $0$ |
| $\tilde{s}_1$ | $a_1$ | $-0.2$ |

Given these definitions $f_t = f(\tilde{r}_0, \ldots, \tilde{r}_t)$. If $f$ admits a representation of the form Equation (14) with $u'$ of constant output dimension, additional state information $h_t$ of the same constant size is possible. Cui & Yu (2023) find a condition of similar form to Equation (14) for the objectives $f$ which their method can optimize:

$$f(\tilde{r}_0, \ldots, \tilde{r}_t) = g\left(\tilde{r}_t, g\left(\tilde{r}_{t-1}, g\left(\tilde{r}_{t-2}, \ldots\right)\right)\right), g : \mathbb{R}^2 \to \mathbb{R}. \tag{17}$$

In this sense, our method can be seen as extending the class of objectives $f$ with constant size extra state information by allowing for $\rho \neq u$ and by allowing a multidimensional update function $u : \mathbb{R}^{k+1} \to \mathbb{R}^k$ instead of $g$. Also, note that the method of Cui & Yu (2023) only works in deterministic environments in conjunction with Q-learning based methods.

### B.2  SUFFICIENT CONDITION FOR CONSTANT SIZE EXTRA STATE INFORMATION $h_t$

While Equation (14) provides a complete categorization of functions with constant size $h_t$, in practice it may be difficult to check whether a given function satisfies this property. In the following, we consider a smaller set of functions including all of the functions in Table 1, and provide an explicit construction of $u$, constant size $h_t$, and $\rho$ for functions of this class. Specifically, we consider function families that can be written with a constant $k \in \mathbb{N}$ as

$$f(\tilde{r}_0, \ldots, \tilde{r}_t) = F\left(t, b_0, \ldots, b_{k-1}\right), \tag{18}$$

where $F : \mathbb{R}^{k+1} \to \mathbb{R}$,

$$b_j = b_j(\tilde{r}_0, \ldots, \tilde{r}_t) = \mathbb{B}_j\left(\varphi_j(0, \tilde{r}_0), \ldots, \varphi_j(t, \tilde{r}_t)\right), \tag{19}$$

where $\varphi_j : \mathbb{R}^2 \to \mathbb{R}$, and $\mathbb{B}_j$ is an arbitrary binary operation, such as $+, \times, \max, \min$, and

$$\mathbb{B}_j(x_0, \ldots, x_t) = \mathbb{B}_j(x_t, \mathbb{B}_j(x_{t-1}, \mathbb{B}_j(\ldots))). \tag{20}$$

For example, if $\mathbb{B}_j$ is the multiplication operation, $\mathbb{B}_j(x_0, \ldots, x_t) = \prod_{i=0}^{t} x_i$. As we show by construction below, all objectives $f$ of this form have extra state information $h_t$ of maximum dimension $k + 1$. Since we are dealing with function families, Equation (18) introduces a notion

Table A3: Objectives $f$ in the form of Equation (18). All $f$ except $\max_{k\in[-1,t]}\sum_{i=0}^{k}\tilde{r}_i$ are permutation invariant resulting in associative $\mathbb{B}_j$.

| $f(\tilde{r}_0,\ldots,\tilde{r}_t)$ | $F(t, b_0,\ldots,b_{k-1})$ | $[\mathbb{B}_j]$ | $[\varphi_j(t,\tilde{r}_t)]$ |
|---|---|---|---|
| $\max(\tilde{r}_0,\ldots,\tilde{r}_t)$ | $h_t^{(0)}$ | $\mathbb{B}_0 = \max$ | $\varphi_0(\tilde{r}_t) = \tilde{r}_t$ |
| $\min(\tilde{r}_0,\ldots,\tilde{r}_t)$ | $h_t^{(0)}$ | $\mathbb{B}_0 = \min$ | $\varphi_0(\tilde{r}_t) = \tilde{r}_t$ |
| Sharpe ratio $\frac{\mathrm{MEAN}(\tilde{r}_0,\ldots,\tilde{r}_t)}{\mathrm{STD}(\tilde{r}_0,\ldots,\tilde{r}_t)}$ | $\frac{h_t^{(0)}/t}{\sqrt{h_t^{(1)}/t - \left(h_t^{(0)}/t\right)^2}}$ | $\mathbb{B}_0 = \mathbb{B}_1 = +$ | $\varphi_0(\tilde{r}_t) = \tilde{r}_t,$ $\varphi_1(\tilde{r}_t) = \tilde{r}_t^2$ |
| $\max_{k\in[-1,t]}\sum_{i=0}^{k}\tilde{r}_i$ | $h_t^{(1)} + h_t^{(0)}$ | $\mathbb{B}_0\left(\varphi_0(\tilde{r}_t), h_t^{(0)}\right) =$ $\max\left(0, h_t^{(0)} - \varphi_0(\tilde{r}_t)\right),$ $\mathbb{B}_1 = +$ | $\varphi_0(\tilde{r}_t) = \tilde{r}_t,$ $\varphi_1(\tilde{r}_t) = \tilde{r}_t$ |
| $\tilde{r}_0\tilde{r}_1\ldots\tilde{r}_t$ | $h_t^{(0)}$ | $\mathbb{B}_0 = \times$ | $\varphi_0(\tilde{r}_t) = \tilde{r}_t$ |
| Harmonic mean $\frac{1}{\frac{1}{\tilde{r}_0}+\cdots+\frac{1}{\tilde{r}_t}}$ | $\frac{1}{h_t^{(0)}}$ | $\mathbb{B}_0 = +$ | $\varphi_0(\tilde{r}_i) = \frac{1}{\tilde{r}_t}$ |
| $\delta^t\sum_{t=0}^{t}\tilde{r}_t,$ $\delta\in(0,1)$ | $\delta^t h_t^{(0)}$ | $\mathbb{B}_0 = +$ | $\varphi_0(\tilde{r}_t) = \tilde{r}_t$ |
| $\frac{1}{t+1}\sum_{k=0}^{t}\tilde{r}_k$ | $\frac{1}{t+1}h_t^{(0)}$ | $\mathbb{B}_0 = +$ | $\varphi_0(\tilde{r}_t) = \tilde{r}_t$ |

of consistency for different sizes of inputs $(\tilde{r}_0,\ldots,\tilde{r}_t)$, since it requires the use of the same $\varphi_j$ for all input sizes. Note that for permutation invariant $f$, $\varphi_j$ is independent of the time-step, i.e. $\varphi_j(i,\tilde{r}_i) = \varphi_j(\tilde{r}_i)$, and $\mathbb{B}_j$ is associative. All but one of the objectives $f$ we consider in this manuscript are permutation invariant.

Given Equation (18), we can directly construct the update function

$$h_{t+1}^{(j)} = u^{(j)}\left(\tilde{r}_t, h_t\right) = \mathbb{B}_j\left(\varphi_j\left(h_t^{(k)}, \tilde{r}_t\right), h_t^{(j)}\right), 0 \le j < k, \tag{21}$$

where superscript $(j)$ indicates the $j$th entry of a vector. Additionally,

$$h_{t+1}^{(k)} = h_t^{(k)} + 1$$

is keeping track of the time step. In this case, each $b_j(\tilde{r}_0,\ldots,\tilde{r}_{t-1})$ is a dimension of $h_t$. Finally,

$$\rho(\tilde{r}_t, h_t) = F\left(h_{t+1}^{(k)}, \left\{h_{t+1}^{(j)}\right\}_{j=0}^{j=k-1}\right) - F\left(h_t^{(k)}, \left\{h_t^{(j)}\right\}_{j=0}^{j=k-1}\right). \tag{22}$$

We show in Table A3 how all functions in Table 1 can be written in this form.

## C  GRID ENVIRONMENT: EXPERIMENTS COMPARING TO CUI & YU (2023)

In this section, we compare our method to the more specialized method for solving NCMDPs introduced by Cui & Yu (2023), which can only be applied in deterministic environments in conjunction with Q-learning methods. We want to answer the question of whether our more general method can be competitive even in this specialized scenario. To this end, we consider the min objective in a grid environment, where each tile is associated with a deterministic reward sampled uniformly from $[-1, 1]$ when initializing the grid. At each step, the agent can choose to move forward and left, forward and right, or forward and straight, and an episode terminates when the agent has crossed from one side of the grid to the other (see Figure A2). We perform experiments on grid sizes $N = 3, 4, 5$ with 10 random grids per size. For each grid and training method, we train 5 agents with different initialization. To facilitate a fair comparison between both methods, we use a vanilla deep Q-learning algorithm with minimal hyperparameters for training (for details see Appendix D). We either extend

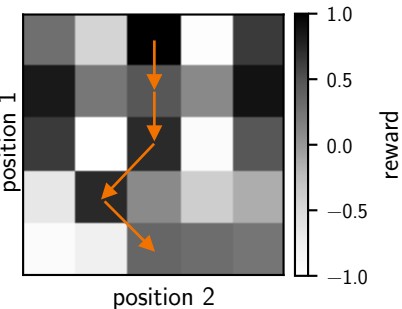

Figure A2: Example of a grid environment for the $\min$ objective with $N = 5$. Orange arrows indicate the ideal policy.

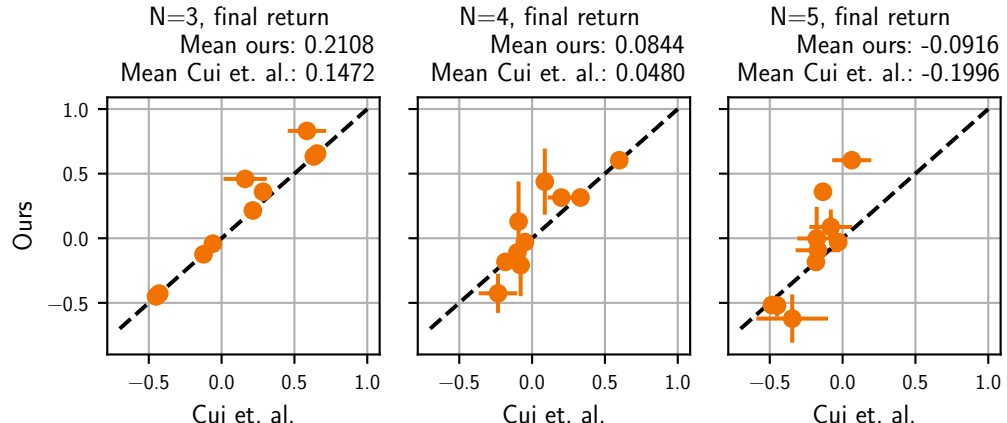

Figure A3: Final return when training agents on the grid environment with our method (y-axis) and the method of Cui & Yu (2023) (x-axis). We perform experiments on grid sizes $N = 3, 4, 5$ with 10 random grids per size (one marker is one grid). Per grid and training method we train 5 agents with different initialization and show the standard deviation as bars. Our method performs better on average for the 3 grid sizes.

the states and adjust the rewards as specified in Table 1 for the minimum objective and use the standard Q-function update of Q-learning (ours), or we only change the Q-function update as specified by Cui & Yu (2023) to $Q(s_t, a_t) \leftarrow \min(r_t, \underset{a}{\mathrm{argmax}}\, Q(s_{t+1}, a))$. We find that our algorithm is not only competitive with the method of Cui & Yu (2023), but even outperforms it in terms of average final performance (see Figure A3) and trains equally fast (see Figure A4). Surprisingly, the prediction loss of the Q-function is similar for both methods (see Figure A5). This suggests that small errors in the Q-function learned by our method may be less detrimental to performance than errors in the Q-function learned by the method of Cui & Yu (2023).

# D  DETAILS ON EXPERIMENTS

**Lunar lander**  For training, we use the PPO implementation of `stables-baselines3` [citepstable-baselines3. The hyperparameters and network architecture of both algorithms were chosen as by Raffin (2020) (which are optimized to give good performance without the velocity penalty), only increasing the batch size and the total training steps to ensure convergence. Training a single agent takes around one hour on a Quadro RTX 6000 GPU with the environment running in parallel on 32 CPUs.

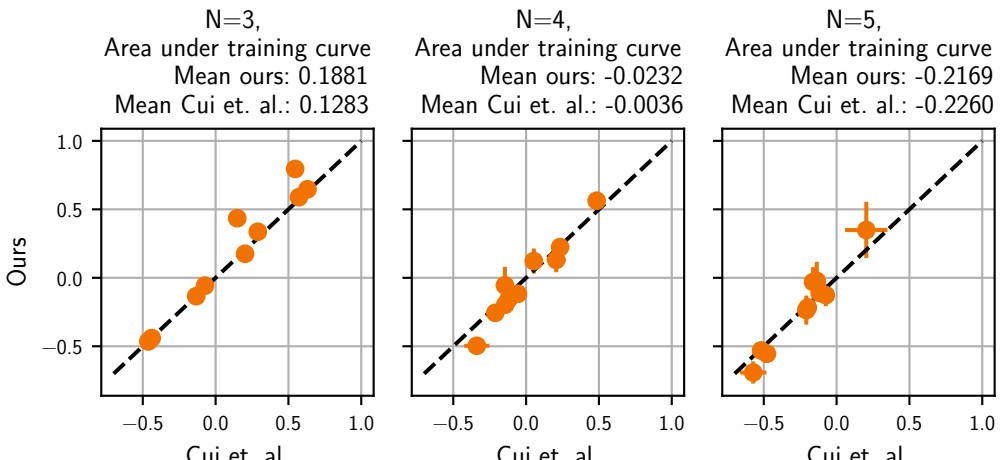

Figure A4: Area under the training curve when training agents on the grid environment with our method (y-axis) and the method of Cui & Yu (2023) (x-axis). We perform experiments on grid sizes $N = 3, 4, 5$ with 10 random grids per size (one marker is one grid). Per grid and training method we train 5 agents with different initialization and show the standard deviation as bars. Both methods perform similarly.

**Portfolio optimization** For training, we use the PPO implementation of `stables-baselines3` (Raffin et al., 2021). The hyperparameters and network architecture of both algorithms were chosen as by Sood et al. (2023) where they were optimized to give a good performance of the DIFFSHARPE algorithm. Training a single agent takes around one hour on a Quadro RTX 6000 GPU with the environment running in parallel on 10 CPUs.

**Peak environment** For training, we use a custom REINFORCE implementation with a tabular policy to keep the number of hyperparameters minimal, updating the agent's policy after each completed trajectory. We use a learning rate of $2^{-10}$ but also performed experiments scanning the learning rate which leaves qualitative results similar. Training a single agent takes around 20 minutes on a Quadro RTX 6000 GPU with the environment running on a single CPU.

**ZX-diagrams** For training, we use the PPO implementation of Nägele & Marquardt (2024) to facilitate the changing observation and action space. The hyperparameters and network architecture were chosen as by Nägele & Marquardt (2024) who originally chose them to give good performance of the PPO algorithm. This is the most compute-intensive experiment reported in this work with one training run lasting for 12 hours using two Quadro RTX 6000 GPUs and 32 CPUs.

**Quantum error correction** For training, we use the PPO implementation of `PureJaxRL` (Lu et al., 2022). For the code discovery task, we modify the reward to consist of the (normalized) difference in immediate rewards used in Olle et al. (2023). The hyperparameters in the code discovery task have been fixed throughout all cases and chosen to provide good performance in all the examples considered with standard PPO. The hyperparameters for the logical state preparation task have been chosen to be the optimal ones reported in Zen et al. (2024), which were optimized for performance of the PPO algorithm. All of the training is done on a single Quadro RTX 6000 GPU. Training 10 agents to complete the code discovery task takes 1 to 2 minutes, depending on the target code parameters. For the logical state preparation task, it takes around 100 seconds for $[[5, 1, 3]]_l$ and around 2500 seconds for $[[17, 1, 5]]_l$ to train 10 agents in parallel.

**Grid environment** For training, we use a vanilla Q-learning algorithm. Per training run, we take a total of $10^5$ steps in the environment. After a warmup phase of 1000 steps, we train after every 100 steps taken in the environment on a batch consisting of the last 1000 experiences of the agent. For exploration, we use an $\epsilon$-greedy strategy with $\epsilon = 0.1$. We use the Adam optimizer with the learning rate linearly annealed from $10^{-2}$ to $10^{-7}$. For the Q-network, we use a multilayer perceptron with 2

hidden layers of dimension 128 with tanh activations. One training run takes around 30 seconds on a single CPU.

We estimate the total compute time to reproduce the results of this manuscript to be around 1000 GPU hours and 25000 CPU hours.

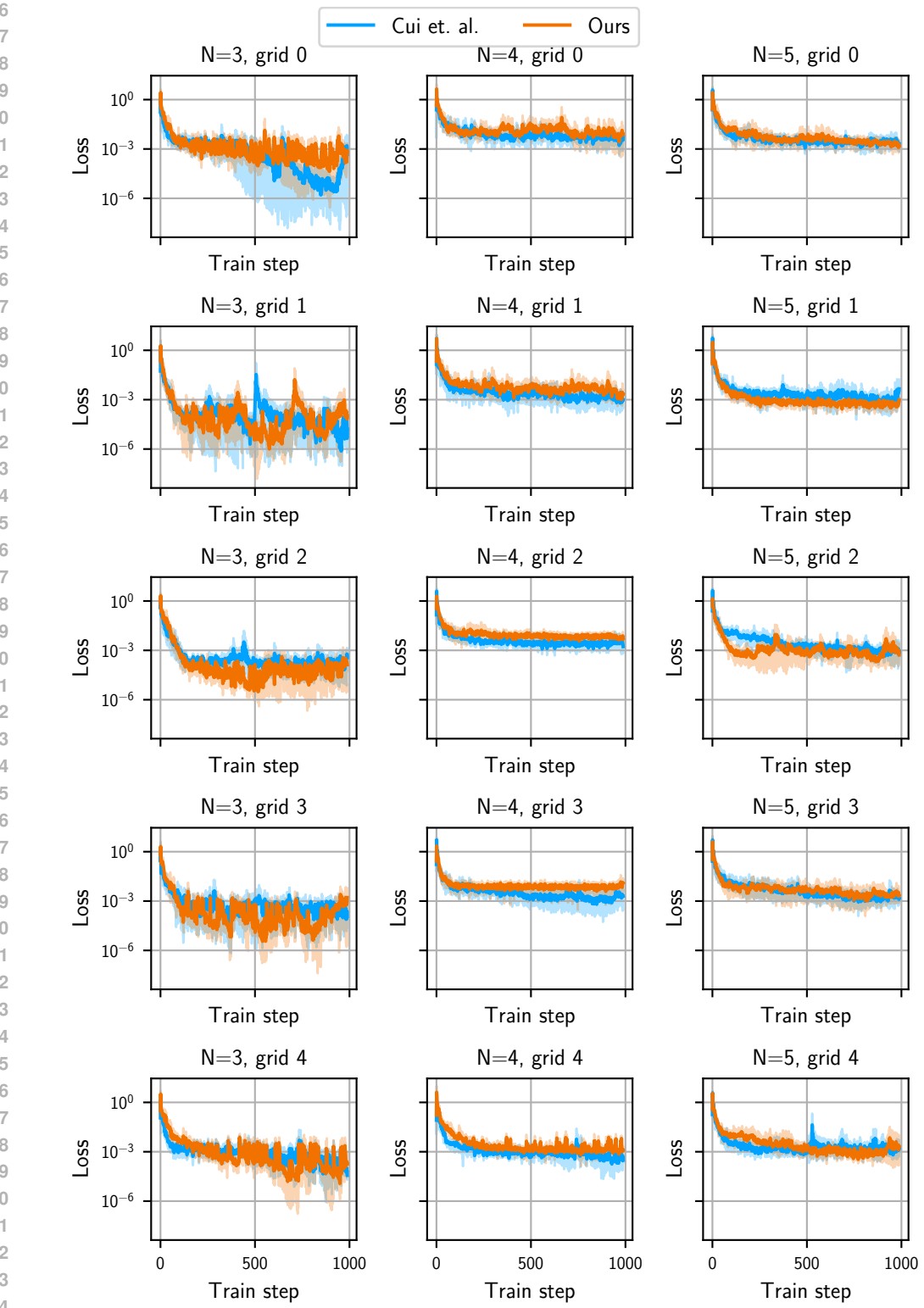

Figure A5: Square error when predicting the Q-function in the grid environment using our method (orange) and the method of Cui & Yu (2023) (blue) against the training step. $N$ indicates the grid size with 10 different random grids per size (only 5 shown) and train 5 agents with different seeds per grid. Lines show the mean loss and the maximum and minimum loss is shaded. The prediction loss of the Q-function is similar for both methods but our method leads to higher performance (see Figure A3).

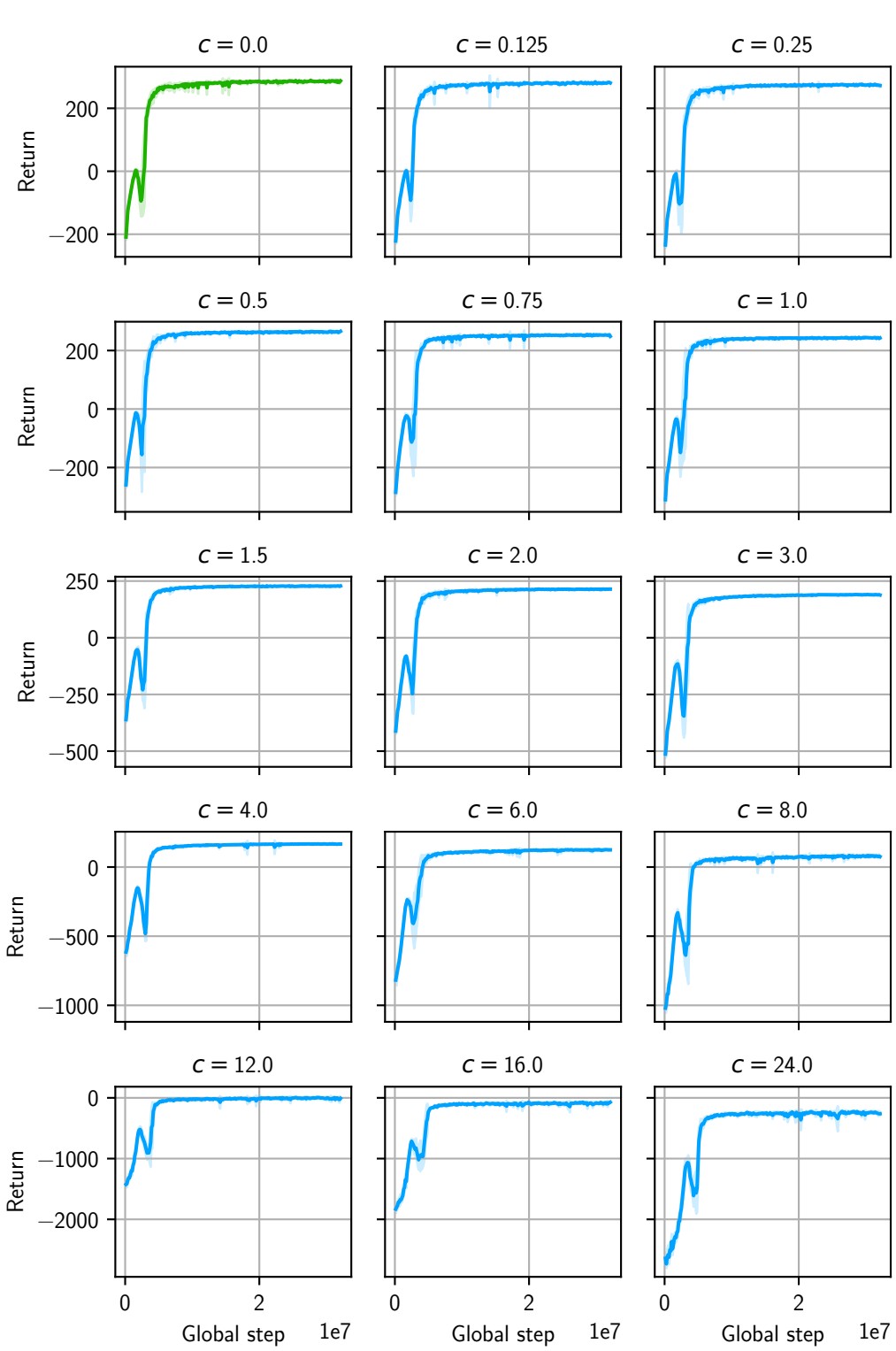

Figure A6: Training progress of cumulative SUMVELPPO algorithm in the Lunar lander environment. The standard deviation over 5 seeds is shaded.

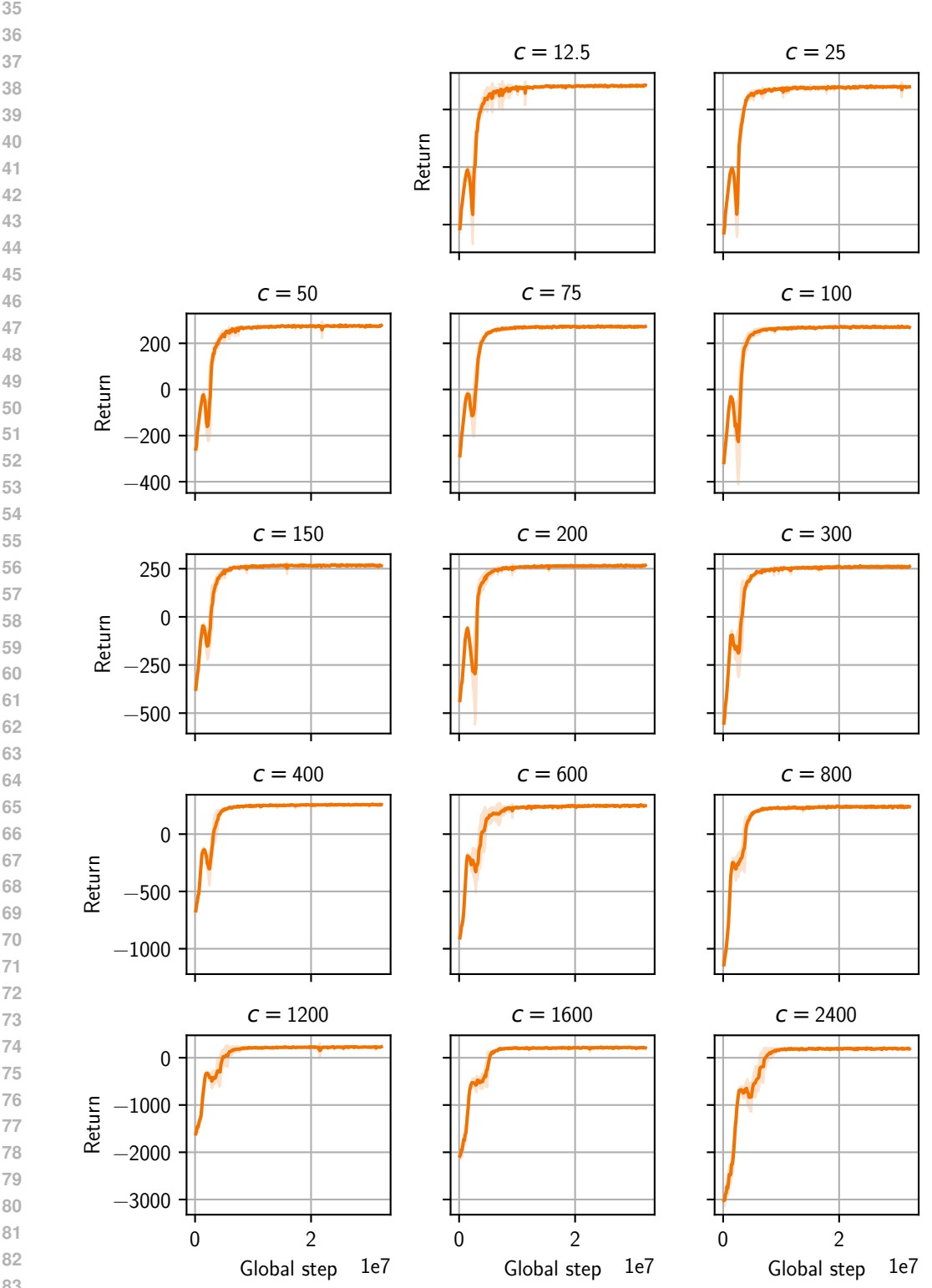

Figure A7: Training progress of non-cumulative MAXVELPPO algorithm in the Lunar lander environment. The standard deviation over 5 seeds is shaded.

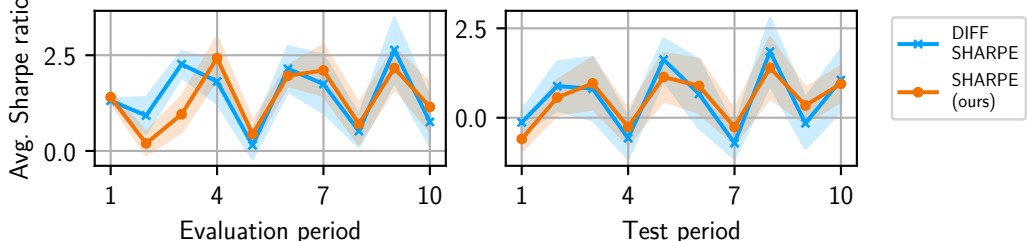

Figure A8: Sharpe ratio on evaluation and test periods.

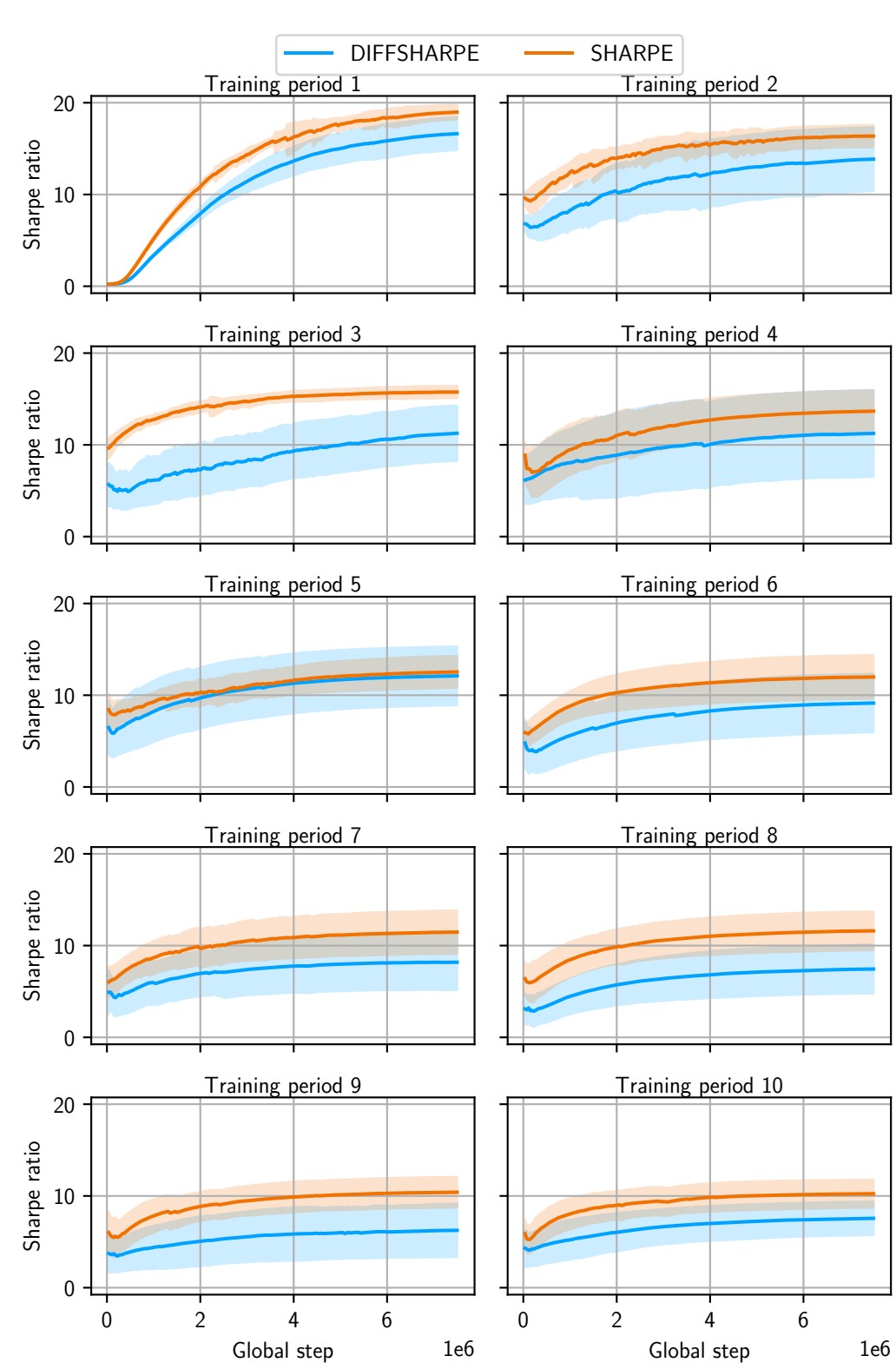

Figure A9: Training progress in the portfolio optimization task. The standard deviation over 25 seeds is shaded.

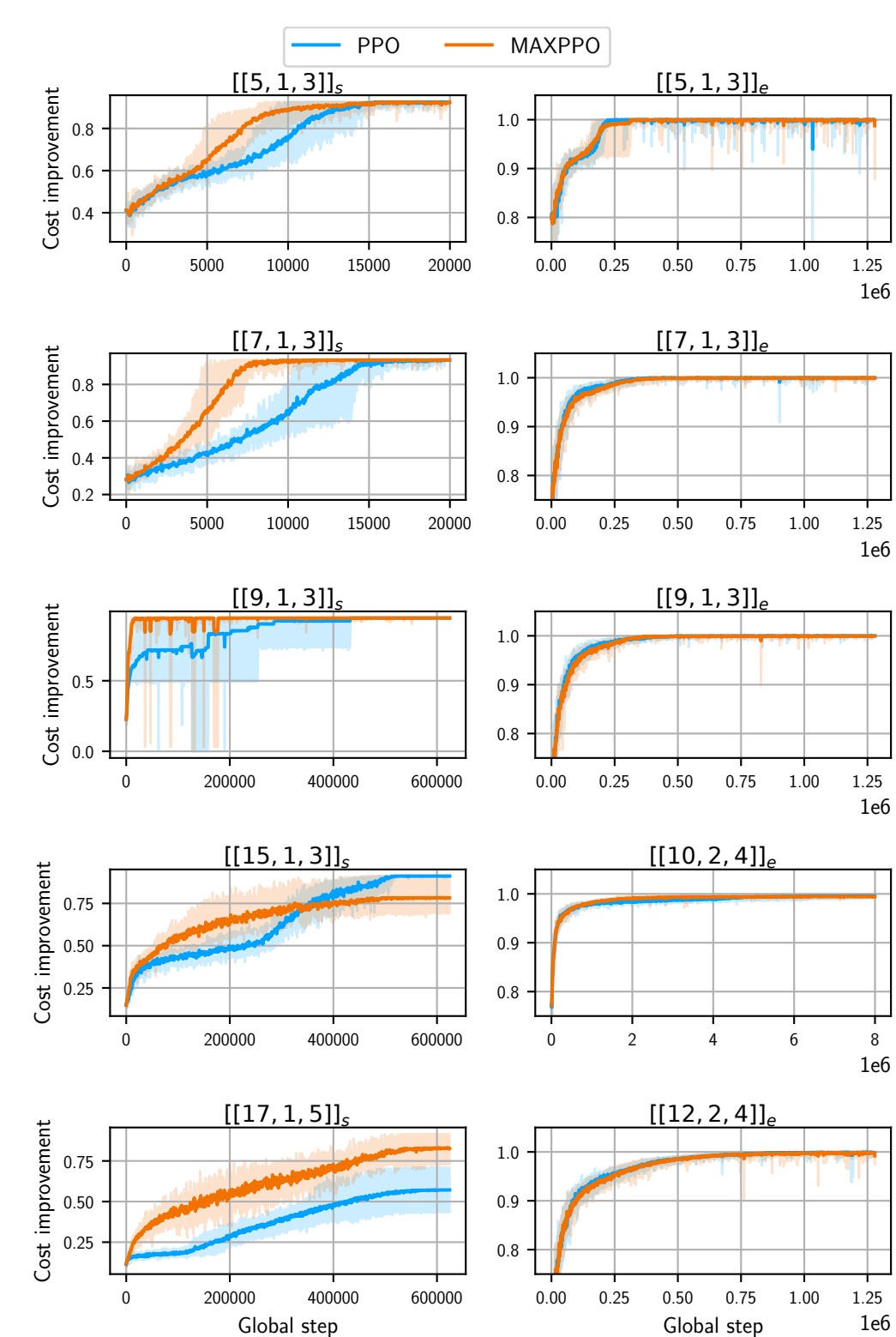

Figure A10: Training progress in the quantum error correction tasks. The best and worst performance of 10 seeds is shaded.

