# OpenReview forum: "Tackling Decision Processes with Non-Cumulative Objectives using Reinforcement Learning"
_ICLR.cc/2025/Conference — Submitted to ICLR 2025_

### Official Review · Reviewer_pdHF · 2024-10-21

**Soundness:** 2
**Presentation:** 4
**Contribution:** 3
**Rating:** 6
**Confidence:** 3

**Summary:**

The authors propose a decision-making framework that allows for the optimization of non-cumulative objectives, by mapping such objectives to the regular cumulative formulation, such that optimizing the cumulative objective results in the optimization of the non-cumulative objective.  The authors show that this MDP framework satisfies important theoretical properties that make it possible to find an optimal policy, and provide several empirical experiments in support of this.

**Strengths:**

This paper provides a theoretically-sound and easy-to-use framework that makes it possible to optimize non-cumulative objectives, by mapping such objectives to the regular cumulative formulation, such that optimizing the cumulative objective results in the optimization of the non-cumulative objective. This framework clearly has several potential use-cases in various domains. The paper is polished, well-written, and the technical concepts are explained in a relatively clear manner.

**Weaknesses:**

I have two concerns regarding this paper:

The first of which is that while the authors do a good job explaining how their framework can be used, they fail to properly formalize the theoretical conditions that they are operating in. For example, they do not address whether the state and action-spaces are or can be finite/infinite, if there is a restriction on the type of policy that can be used (stationary, stochastic, etc.), as well as whether there are implications of their framework being used in off-policy settings. Because of this, the extent to which their framework can be utilized is unclear.

My second concern is related to the experiments. In particular, the comparison of non-cumulative objectives to the cumulative objects seems unintuitive, given that no-where in the paper do the authors claim the non-cumulative objectives are better than the cumulative objectives. Rather, it would be more intuitive, and consistent with the rest of the text, if the experiments showed that the proposed methods indeed are able to find the optimal policy for the non-cumulative objective. As such, it would appear that the experiments in the current draft of the paper do little to support the claim that the proposed methods can be used to optimize non-cumulative objectives.

As such, I am open to increasing my score if the authors can properly formalize the theoretical conditions that they are operating in, and either 1) the authors can convince me that the experiments in the current draft are adequate, or 2) the authors improve the existing experiments in such a way that they address my concerns, or 3) the authors can provide additional experiments that address my concerns.

**Questions:**

1) What are the off-policy implications of this method?
2) A regular average would appear to be a common and useful non-cumulative objective to include in the paper. Was there a reason that it was not included?

---

> ### Author Response · Authors · 2024-11-19
>
> We thank the reviewer for their time and valuable feedback. We now comment on the points raised:
>
> **[...]they do not address whether the state and action-spaces are or can be finite/infinite, if there is a restriction on the type of policy that can be used (stationary, stochastic, etc.), as well as whether there are implications of their framework being used in off-policy settings. Because of this, the extent to which their framework can be utilized is unclear.**
>
> Our framework maps a general NCMDP to a corresponding MDP. Then, any standard method can be used to solve the corresponding MDP. Thus, our framework is completely flexible and agnostic to the type of action space or employed policy.
> The only limitation we state in the 'Preliminaries' is that we restrict ourselves to an episodic setting (although this could potentially be extended, see reply to reviewer cuNK).
>
> 1) The choice of action space is therefore completely arbitrary. In particular, finite, infinite, discrete, and continuous action spaces are supported.
> 2) The type of policy depends on which solver is chosen to solve the corresponding MDP. In this sense, any type of policy is possible as long as it accommodates the action space of the corresponding MDP (which is the same as the action space of the underlying NCMDP).
> 3) Our framework can be used in off-policy settings, as long as an off-policy algorithm is chosen to solve the corresponding MDP.
>
> We now emphasize this in the conclusion by adding 'As our method is agnostic to the algorithm used to solve the resulting MDP, it works for arbitrary action spaces and in conjunction with both off- and on-policy algorithms.'
>
> **'My second concern is related to the experiments. In particular, the comparison of non-cumulative objectives to the cumulative objects seems unintuitive, given that no-where in the paper do the authors claim the non-cumulative objectives are better than the cumulative objectives. Rather, it would be more intuitive, and consistent with the rest of the text, if the experiments showed that the proposed methods indeed are able to find the optimal policy for the non-cumulative objective. As such, it would appear that the experiments in the current draft of the paper do little to support the claim that the proposed methods can be used to optimize non-cumulative objectives.'**
>
> The experiments we present are examples of real world problems with a non-cumulative objective that is established and clearly defined by the respective research communities (e.g. maximize the Sharpe ratio in finance, find the state with lowest cost function reachable from the inital state in discrete optimization). However, in the absence of a method to solve general NCMDPs, these real-world problems have previously been tackled by standard MDPs that can only approximate the true objective (differential Sharpe ratio approximates the non-cumulative Sharpe ratio in finance, sum of cost differences approximates the maximum cost reduction in discrete optimization problems). As such, these previous state-of-the-art methods provide a reasonable baseline to compare our method to.
>
>
> There are also two other types of experiments in the manuscript:
> 1) As suggested by the reviewer: Apply our method in a simple setting where the optimal policy is known and show that our method finds the optimal policy.
>
>     We show such a proof-of-principle experiment in Figure 2, and demonstrate that our method (paired with dynamic programming as an MDP solver) can successfully find the (in this case known) optimal policy (see Figure A1, and Table A1). We believe that the theoretical foundation of our work is strong enough that further empirical demonstrations on such toy examples are unnecessary.
>
> 2) In the new Appendix C, we now also compare our method with the only previous method for solving NCMDPs with objectives other than the max function: The work of Cui et. al. (doi: 10.1109/TMLCN.2023.3285543), which is only applicable in deterministic settings in conjunction with Q-learning based methods and a more restrictive class of functions $f$ than our method. We find that even in this restricted setting, our more general method slightly outperforms the method of Cui et. al. (see Appendix C for details).
>
>
> **'What are the off-policy implications of this method?'**
>
> See our answer above: Our method is compatible with off-policy methods, since the choice of MDP solver is completely flexible.
>
> **'A regular average would appear to be a common and useful non-cumulative objective to include in the paper. Was there a reason that it was not included?'**
>
> Thanks for pointing this out. There was no specific reason. In fact, we now include this objective in Table 1. It could be interesting in settings where the agent can influence the trajectory length by its actions.

---

> > ### Comment · Reviewer_pdHF · 2024-11-20
> >
> > I thank the authors for their reponse and clarifications. It is impressive that the proposed method can be applied in the off policy case. Overall, the proposed method has clear merit and utility, so I have revised my score. However, I stress to the authors that without a compelling empirical result (none of the results in the updated draft are particularly compelling), the paper's impact will be severly diminished. Hence, I strongly encourage the authors to include a non-trivial result that shows that the proposed method is able to indeed find the optimal policy for a non-cumulative objective.

---

> > > ### Author Response · Authors · 2024-11-26
> > >
> > > We thank the reviewer for their comment and for revising their score. The reviewer is interested in a **'non-trivial result that shows that the proposed method is able to indeed find the optimal policy for a non-cumulative objective'**.
> > >
> > > The proof-of-principle experiment in Figure 2 is by no means trivial. This experiment captures the key conceptual difficulties in solving NCMDPs: A non-cumulative reward function in a stochastic environment. We show that our method can successfully find the optimal policy in this case (Figure A1, Table A1). Note that **no previous method for solving NCMDPs could solve this example** (Cui. et al. can maximize the minimum reward but not in stochastic environments, other methods cannot even maximize the minimum reward in any setting). **As a new comparison, we now show that the method of Cui et. al. indeed fails to find the optimal policy in this example, while our method is succesful (see new Table A2)**.
> > >
> > > We now also more clearly motivate the choice of our experiments in the beginning of Section 3.
> > >
> > > Could the reviewer clarify which specific other experiments the reviewer would find more compelling?

---

> > > > ### Comment · Reviewer_pdHF · 2024-11-26
> > > >
> > > > I thank the authors for their most recent comments. I recognize that the line between "compelling" and "not compelling" is an arbitrary one, so I will try to clarify what I mean. My worry is that only showing that the proposed approach can find the optimal policy on a two-step decision process may give the wrong impression to an arbitrary reader that the proposed method can only (optimally) handle simple scenarios like the two-step decision process. The paper would hence greatly benefit from having a more complex experiment, where the authors show that their proposed approach can optimally handle, for example, large state and action-spaces, function approximation, etc. Having such an experiment would get the point across to the reader that the optimality of the proposed approach is not limited to simple toy experiments. Remember that most readers are skeptical; they will assume that if something is not shown, it is because there is a limitation that the authors are trying to hide. Hence, the paper needs to have some compelling result which shows that (the optimality of) the proposed approach can be applied universally, beyond toy experiments.
> > > >
> > > > I hope that this clarifies my previous comment.

---

> > > > > ### Author Response · Authors · 2024-12-03
> > > > >
> > > > > We would like to thank the reviewer again for the overall useful feedback. However, we disagree on this point. We do not think that a larger toy example ("large state and action spaces, function approximation") would benefit our work. Since the reviewer is not interested in showing just good performance, but optimal performance, all such examples are limited to settings where the optimal policy is known. Spending a large amount of computational resources on these already solved problems seems excessive to us.
> > > > > Moreover, in real world settings, reinforcement learning with function approximation is commonly applied to problems that are so complex that the ideal policy is unknown and likely unattainable. The goal here is not to show that optimal performance is achieved, but to show practical improvements over previous methods.
> > > > > We focus on such problems (including large state and action spaces and function approximation) in the experimental part and show improvements over the previous state of the art in a range of diverse problems.

---

### Official Review · Reviewer_cuNK · 2024-10-26

**Soundness:** 2
**Presentation:** 3
**Contribution:** 2
**Rating:** 6
**Confidence:** 4

**Summary:**

In this paper, the authors consider the problem of control in finite-horizon Non-cumulative MDPs (NCMDPS). They propose a general mapping of NCMDPs to standard MDP, and hence the problem can be solved by existing standard RL methods. They use a series of numerical experiments to illustrate the NCMDP problems and the proposed transformations.

**Strengths:**

* The authors consider the problem of control in NCMDPs, which is important and understudied in the RL community.

* The proposed mapping of NCMDPs to standard MDPs is simple and intuitive, and can effectively solve many problem instances of NCMDPs.

* The empirical studies are concrete, showing both the necessity of considering the problem of control in NCMDPs and the effectiveness of the proposed mappings.

**Weaknesses:**

* The paper has technical flaws, i.e., the function $f$ is ill-defined. In the original problem statement, $f$ is defined as a function on $\mathbb{R}^T$. But later on, the authors also use notations like $f(r_1,...,r_t)$, where $f$ should be treated as a function on $\mathbb{R}^t$. I think it may be helpful to define $f$ as a function of a set ($f(\\{r_1,...,r_t\\})$, $\\{r_1,...,r_t\\}$ is the set of $t$ rewards, $t$ can be any integer between $1$ and $T$). This works for all problem instances in Table 1.

* The authors overstate their contributions. They claim their methods work for arbitrary functions $f$. However, as indicated in the previous comment, Definition 1 is invalid for general $f$ on $\mathbb{R}^T$. Even if $f$ is treated as a function of a set, it turns out only for certain choices of $f$ there exist fix-size $h_t$ and corresponding functions $u$ and $\rho$.  I think the authors should try to find a classification of functions f with constant-size state adaptions. Even some preliminary results such as sufficiency conditions or necessary conditions would be helpful.

* I think in empirical studies the authors should compare their methods with prior works on the control of NCMDPs with a specific choice of objectives and I am curious if the proposed method (which is more general) can produce comparable performances.

**Questions:**

* Can the proposed mapping of NCMDPs to MDPs be generalized to infinite-horizon MDPs? (when $f$ is defined as a function of set.)

* It seems that if we transformed an NCMDP to a standard MDP, the resulting standard MDP is governed by the dynamics of $s_t$ and $h_t$ can be viewed as an auxiliary state. I am curious whether there are some connections with the framework called MDPs with latent dynamics (see [1], [2]).

[1] Simon Du, Akshay Krishnamurthy, Nan Jiang, Alekh Agarwal, Miroslav Dudik, and John Langford. Provably
efficient RL with rich observations via latent state decoding. In International Conference on Machine
Learning, 2019.

[2] Amortila, Philip, et al. "Reinforcement Learning under Latent Dynamics: Toward Statistical and Algorithmic Modularity." arXiv preprint arXiv:2410.17904 (2024).

---

> ### Author Response · Authors · 2024-11-19
>
> We thank the reviewer for their time and valuable feedback.
>
> **'[...] $f$ is defined as a function on $\\mathbb{R}^T$. But later on, the authors also use notations like $f(r_0, \\dots, r_t)$, where $f$ should be treated as a function on $\\mathbb{R}^t$. I think it may be helpful to define $f$ as a function of a set.'**
>
> We thank the reviewer for pointing out this source of possible confusion and agree that $f$ should be defined more rigorously. However, set functions are not applicable, since sets have no notion of order and do not support duplicate elements. Thus, we define $f$ as a family of functions and add after Equation (2):
> '[...], we require $f$ to be a family of functions consisting of a function $f_t : \\mathbb{R}^t \\rightarrow \\mathbb{R}$ for each $t \\in \\mathbb{N}$. For brevity, we denote $f_t(\\tilde r_0, \\tilde r_1, \\dots, \\tilde r_{t-1}) = f(\\tilde r_0, \\tilde r_1, \\dots, \\tilde r_{t-1})$.'
>
> **'The authors should try to find a classification of functions f with constant-size state adaptions. Even some preliminary results [...] would be helpful.'**
>
> We now discuss conditions on $f$ to allow for constant size state adaptions $h_t$ in the new Appendix B:
> 1) We provide a necessary and sufficient condition for objectives $f$ with constant size additional state information $h_t$ (see Appendix B.1).
> 2) We also provide a sufficient, but not necessary, condition that is easier to verify and covers a large class of objectives, e.g. all functions in Table 1. Specifically, we consider function families that can be written with a constant $k\in\mathbb{N}$ as
> $$
> f(\\tilde r_0, \\dots, \\tilde r_t) = F\\left(t, b_0, \\dots, b_{k-1}\right),
> $$
> where $F: \\mathbb{R}^{k+1} \\rightarrow \\mathbb{R}$,
> $$
>     b_j = b_j(\\tilde r_0, \\dots, \\tilde r_t) =B_j\\big(\\varphi_j(0, \\tilde r_0), \\dots, \\varphi_j(t, \\tilde r_t)\\big),
> $$
> where $\\varphi_j : \\mathbb{R}^2 \\rightarrow \\mathbb{R}$, and $B_j$ is an arbitrary binary operation, such as $+, \\times, \\max, \\min$, and
> $$
>    B_j(x_0, \\dots, x_t) = B_j(x_t, B_j(x_{t-1}, B_j(...))).
> $$
> For details and an explicit construction of $u$, $\\rho$, and $h_t$ with dimension $k+1$, see the new Appendix B.2.
>
>
> **'They claim their methods work for arbitrary functions $f$. However, [...] it turns out only for certain choices of $f$ there exist fix-size $h_t$'**
>
> Our method works for arbitrary functions by appending all past rewards to the state. This is a realistic possibility as the state space grows exponentially with the trajectory length, but the representation of a single state grows only linearly and reinforcement learning methods have successfully been applied to a variety of problems with inherently exponentially large state spaces (e.g. Chess, Go). Of course, finding constant size $h_t$ can improve the efficiency of our method.
>
>
> **'The authors should compare their methods with prior works on the control of NCMDPs with a specific choice of objectives and I am curious if the proposed method (which is more general) can produce comparable performances.'**
>
> In the new Appendix C, we compare our method with the only previous method for solving NCMDPs with objectives other than the max function: The work of Cui et. al. (doi: 10.1109/TMLCN.2023.3285543). Specifically, we consider the problem of traversing a grid with deterministic rewards on each tile together with the minimum objective. We find that even in this restricted setting, our more general method slightly outperforms the method of Cui et. al. (see Appendix C for details).
>
> **'Can the proposed mapping of NCMDPs to MDPs be generalized to infinite-horizon MDPs? (when is defined as a function of set.)'**
>
> We believe that any algorithm used to solve infinite-horizon MDPs could, in principle, be used to solve the corresponding MDP of an infinite-horizon NCMDP. However, a full proof would require checking the convergence of the now infinite sums in the proof of Theorem 1.
> For example, optimizing the average Sharpe ratio in an infinite horizon NCMDP seems like a reasonable application.
>
> **'[...]I am curious whether there are some connections with the framework called MDPs with latent dynamics (see [1], [2]).'**
>
> We are not sure whether the following answers your question. Please clarify, if not.
>
> In the following, we understand MDPs with latent dynamics as in source [1] to be 'episodic MDPs with rich observations generated from a small number of latent states'.
> Consider an NCMDP with a corresponding MDP where we append all previous rewards to the state, i.e.,  $h_t = [\tilde r_0, \dots, \tilde r_{t-1}]$. Consider also a second MDP corresponding to the same NCMDP with a lower dimensional extra state information $h_t^\prime$ (e.g. $h_t^\prime \in \mathbb{R}$). In the context of MDPs with latent dynamics, $h_t$ can now be seen as part of a 'rich observation' (i.e. an observation containing redundant information) governed by the latent variable $h_t^\prime$.

---

> > ### Comment · Reviewer_cuNK · 2024-11-22
> >
> > Thanks for the response. Since the authors' response addresses most of my concerns, I will increase my rating to 6.

---

### Official Review · Reviewer_4ejM · 2024-10-31

**Soundness:** 2
**Presentation:** 2
**Contribution:** 2
**Rating:** 3
**Confidence:** 3

**Summary:**

This paper studies the problem of finding an optimal policy for a special class of decision problem called non-cumulative Markov decision process (NCMDPs) where instead of the sum of rewards, it aims to maximize the expected value of an arbitrary function of the rewards. The paper solves NCMDPs by mapping it to standard MDPs, allowing direct application of MDP solver for NCMDPs. It also performs numerical experiments of classical control, portfolio optimization, and discrete optimization that use NCMDP objectives, and shows that their method improves both training time and final performance compared with standard MDP solvers with cumulative reward.

**Strengths:**

- The NCMDP setting considered in this paper fits many applications that does not directly fit into the MDP setting with cumulative rewards. Some examples include the weakest-link problem in network routing which maximizes minimum reward, the Sharpe ratio in finance which maximizes the mean divided by standard deviation.
- The paper provides a straightforward solution to NCMDPs by first map an NCMDP to a standard MDP, which allows direct application of black box MDP solvers.
- It provides comprehensive empirical results applying this method to a range of environments with non-cumulative reward objective, including classical control, portfolio optimization with Sharpe ratio, discrete optimization with lost cost objective.

**Weaknesses:**

- The mapping from NCMDP to MDP provided in equations (3)-(5) augments the state with $h_t$ that represent necessary information for reward history, and is updated as $h_{t+1}=u(h_t,\tilde{r}_t)$ and satisfies $r_t=\rho(h_t,\tilde{r}_t)$. For arbitrary reward function $f(r)$, an essential factor to ensure the mapping to MDP is of reasonable size is to find an efficient functional form of $u$ and $\rho$ that summarizes this information from reward history. However, the paper only shows a list of examples of special examples of $f$ and does not specify how to construct $u$ and $\rho$ for arbitrary function $f$ in general, and does not provide an bound to the resulting state dimensions, which at worst case must contain the set of all historical rewards. I think this makes the overall contribution of the paper limited.
- In the experimental section, for each environment, the paper compares their method of MDP reduction (with modified state and reward) with an RL agent with cumulative objective. This does not seem like a very fair comparison given the cumulative objective will be the wrong objective in each of these environments, and it is clear that correct specification of the objective will lead to better performance.

**Questions:**

- The paper gives very specific examples of construction of $u$ and $\rho$ that allows an efficient reduction from NCMDP to MDP of small augmented state size. Can you give examples of more general classes of non-cumulative objective function $f$ that induces $h_t$ that is of reasonably small dimensions?

---

> ### Author Response · Authors · 2024-11-19
>
> We thank the reviewer for their time and valuable feedback. We now comment on the points raised:
>
> **'[...] The paper gives very specific examples of construction of $u$ and $\rho$ that allows an efficient reduction from NCMDP to MDP of small augmented state size. Can you give examples of more general classes of non-cumulative objective function that induces $h_t$ that is of reasonably small dimensions?'**
>
> We now discuss conditions on $f$ to allow for constant size extra state information $h_t$ in the new Appendix B:
> 1) We provide a necessary and sufficient condition for objectives $f$ with constant size additional state information $h_t$. Namely, that we can write
> $$f(\\tilde r_0, \\dots,\ \tilde r_t)  = \\rho^\\prime\\left(\\tilde r_t, u^\prime\\left(\tilde r_{t-1}, u^\\prime\left(\\tilde r_{t-2}, ...\\right)\\right)\\right),$$
> with $\rho^\\prime: \\mathbb{R}^{k+1} \\rightarrow \\mathbb{R}$ and $u^\\prime: \\mathbb{R}^{k} \\rightarrow \\mathbb{R}$, where $k+1 \\in \\mathbb{N}$ is the dimension of $h_t$. This can be seen as an extension of the optimal substructure condition used by Cui et. al. (doi: 10.1109/TMLCN.2023.3285543).
> 2) We also provide a sufficient, but not necessary, condition that is easier to verify and covers a large class of objectives, e.g. all functions in Table 1. Specifically, we consider function families that can be written with a constant $k\\in\\mathbb{N}$ as
> $$
> f(\\tilde r_0, \\dots, \\tilde r_t) = F\\left(t, b_0, \\dots, b_{k-1}\\right),
> $$
> where $F: \\mathbb{R}^{k+1} \\rightarrow \\mathbb{R}$,
> $$
>     b_j = b_j(\\tilde r_0, \\dots, \\tilde r_t) = B_j\\big(\\varphi_j(0, \\tilde r_0), \\dots, \\varphi_j(t, \\tilde r_t)\\big),
> $$
> where $\\varphi_j : \\mathbb{R}^2 \\rightarrow \\mathbb{R}$, and $B_j$ is an arbitrary binary operation, such as $+, \times, \max, \min$, and
> $$
>    B_j(x_0, \\dots, x_t) = B_j(x_t, B_j(x_{t-1}, B_j(...))).
> $$
> For example, if $B_j$ is the multiplication operation, $B_j(x_0, \\dots, x_t) = \\prod_{i=0}^{t} x_i$. For details on the explicit construction of $u$, $\\rho$, and $h_t$ with dimension $k+1$, see the new Appendix B.2.
>
> **'[...] the paper only shows a list of examples of special examples of and does not specify how to construct and for arbitrary function in general, and does not provide an bound to the resulting state dimensions, which at worst case must contain the set of all historical rewards.'**
>
> We agree that this is a limitation of our work. However, we partially resolved it (see previous answer). Moreover:
>
>
> 1) When appending all past rewards to the state, the state space grows exponentially with the trajectory length, but the representation of a single state grows only linearly. Reinforcement learning methods have succesfully been successfully applied to a variety of problems with inherently exponentially large state spaces, but modest trajectory lengths (e.g. Chess, Go). Therefore, appending all past rewards to the state may not be limiting in some problems (though it may be in others).
> 2) Even only the functions specified in Table 1 are of interest to large reinforcement learning communities (finance, discrete optimization).
>
>
>
> **'[...] the paper compares their method of MDP reduction (with modified state and reward) with an RL agent with cumulative objective. This does not seem like a very fair comparison given the cumulative objective will be the wrong objective in each of these environments, and it is clear that correct specification of the objective will lead to better performance.'**
>
> We agree, and the fact that, using our method, it is now possible to specify the correct objective to an RL algorithm is a main advancement of our paper:
> 1) In the absence of a method for solving general NCMDPs, the real-world problems we consider in the experiment section have so far been solved by standard MDPs that only approximate the true objective (differential Sharpe ratio approximating non-cumulative Sharpe ratio in finance, sum of cost differences approximating maximum cost reduction in discrete optimization problems). As such, these previous state-of-the-art methods provide a reasonable baseline against to compare our method.
>
> 2) We also show a proof-of-principle experiment in Figure 2, and demonstrate that our method (paired with dynamic programming as an MDP solver) can successfully find the (in this case known) optimal policy (see Figure A1, and Table A1).
>
> 3) In the new Appendix C, we now also compare our method with the only previous method for solving NCMDPs with objectives other than the max function: The work of Cui et. al. (doi: 10.1109/TMLCN.2023.3285543), which is only applicable in deterministic settings in conjunction with Q-learning based methods and a more restrictive class of functions $f$ than our method. We find that even in this restricted setting, our more general method slightly outperforms the method of Cui et. al. (see Appendix C for details).

---

> > ### Comment · Reviewer_4ejM · 2024-11-25
> >
> > I thank the authors for their responses, and the updated Appendix B does help me understand more rigorously the generality of the methods. The proposed method has utilities in handling MDPs with non-cumulative dynamics given the conditions specified by Appendix B, and the additional empirical result in Appendix C shows advantage compared with baseline algorithms. However, I am a bit hesitant to raise my score higher given the limitation of function classes this method can apply to and the limitations of comparisons in the empirical section.

---

> > > ### Author Response · Authors · 2024-11-26
> > >
> > > We thank the reviewer for their comment and are pleased that our new appendix has helped to clarify some unclear points.
> > >
> > > The reviewer is concerned about the **'comparisons in the empirical section'**.
> > > We clarify the choice of comparisons in the following. We present three types of experiments:
> > > 1. A proof-of-principle experiment in Figure 2. This experiment captures the main conceptual difficulties of solving NCMDPs: A non-cumulative reward function in a stochastic environment. We show that our method (coupled with dynamic programming as the MDP solver) can successfully find the (in this case known) optimal policy (see Figure A1 and Table A1). Note that no previous method could solve this example (Cui. et al. can maximize the minimum reward, but not in stochastic environments, other methods cannot even maximize the minimum reward in any setting). **As a new comparison, we now show that the method of Cui et. al. indeed fails to find the optimal policy in this example, while our method is successful (see new Table A2)**.
> > > 2. We show applications to **real-world tasks** that have previously been tackled by reinforcement learning. However, previous methods were severely limited because they could not specify the true (non-cumulative) objective to the RL agent. Thus, previous state-of-the-art approaches relied on standard MDPs that only approximate the true objective (differential Sharpe ratio approximates non-cumulative Sharpe ratio in finance, sum of cost differences approximates maximum cost reduction in discrete optimization problems). These approximate methods are the state of the art and therefore a suitable comparison for our approach. **We show that by specifying the true non-cumulative objective to the agent, performance in these real-world tasks can be improved as compared to the previous state-of-the-art.**
> > > 3. A comparison of our method with the only previous method for solving NCMDPs with objectives other than the max function.  We find that **even in the highly restricted setting in which this method can be applied, our more general method performs better.**
> > >
> > > In summary, **we have compared our algorithm in three different settings to the best available baseline and showed performance improvements in all of them**.
> > >
> > > We have updated the manuscript to make the motivation for the experiments and the chosen baselines clearer (beginning of Section 3) and are interested in what specific other comparisons the reviewer would be interested in?
> > >
> > > The reviewer is also concerned about **'the limitation of function classes this method can apply to'**.
> > > 1) **The function class that results in constant size extra state information is much larger than the function classes that could be maximized by any of the previous methods:** In fact, only the maximum function can be maximized in stochastic environments by previous methods (Veviurko et. al. ICML 2024). For some other functions in Table 1, only the method of Cui. et. al. is available, which works only in conjunction with Q-learning in deterministic settings. Finally, the functions in the third and fourth row of Table 1 cannot be maximized by any previous method. Empirically, **we could not find a single interesting non-cumulative objective outside of the class specified in Appendix B2**. We are interested in which interesting objectives the reviewer expects to fall outside of this class?
> > > 3) Our method is applicable to general function classes (by appending all past rewards to the state, see comment above).

---

### Official Review · Reviewer_ivnZ · 2024-11-06

**Soundness:** 3
**Presentation:** 3
**Contribution:** 2
**Rating:** 5
**Confidence:** 4

**Summary:**

In traditional MDPs, the agent is tasked with the optimisation of the policy value function that is the expected sum of the rewards. In a non cumulative markov decision process, the value function is replaced by the expected value of an arbitrary function of the rewards.
The authors propose a method to translate an NCMDP into a regular MDP and show the expected returns of the two are equal and apply MDP solving methods to NCMDPs.

**Strengths:**

Definition 1 and the possible applications of the method are interesting.

**Weaknesses:**

The variables u, h and $\rho$ are discussed in the paragraph following the statements of equations 3,4,5. You should introduce them before.
Also, their description is too vague. For example "This can be achieved by extending the state space with ht , which preserves all necessary
information about the reward history" does not give me a good sense of what the function u should be.
I suggest the statement of theorem 1 be rearranged to: assumptions then conclusion, instead of the current: assumption then conclusion then more assumptions.

**Questions:**

It seems to me that if the update function u of definition 1 is a constant function, no information is preserved and the constructed MDP could not have the same value function as the NCMDP. Is this right? If it is, this sort of choice should be excluded in definition 1.
Later in the same paragraph, the authors note that an h_t recording all reward history is sufficient to construct an MDP but the state space is exponentially large so there must be some tradeoff between state space and "Markovness". How much information can be lost from the full history before we lose the Markov condition? Is there a characterisation of the functions u that satisfy this property?

---

> ### Author Response · Authors · 2024-11-19
>
> We thank the reviewer for their time and valuable feedback. We now comment on the points raised:
>
> **'It seems to me that if the update function u of definition 1 is a constant function, no information is preserved and the constructed MDP could not have the same value function as the NCMDP. Is this right? If it is, this sort of choice should be excluded in definition 1.'**
>
> We think there may be a slight misunderstanding: We only claim that if Definition 1 is satisfied, Theorem 1 holds. However, Definition 1 is not generally satisfied for all $u$ and $\rho$.
> To your comment: If $u$ is chosen as a constant function, then $h_t$ is also constant, and Eq. (3), i.e., $r_t = \rho(\tilde r_t, h_t)$ cannot be satisfied for any $\rho$. Therefore, a constant $u$ is already excluded by Definition 1.
>
> **'Later in the same paragraph, the authors note that an h_t recording all reward history is sufficient to construct an MDP but the state space is exponentially large so there must be some tradeoff between state space and "Markovness". How much information can be lost from the full history before we lose the Markov condition? Is there a characterisation of the functions u that satisfy this property?'**
>
> 1) All functions $u$ satisfying Definition 1 preserve full 'Markovness', so our scheme always chooses 'Markovness' in the trade-off between 'Markovness' and size of the state space.
> 2) It is an interesting question to ask which functions $f$ allow (small) constant size extra state information $h_t$. We now discuss some results related to this in Appendix B: Specifically, we provide a necessary and sufficient condition for objectives $f$ with constant size additional state information $h_t$. However, this condition might be difficult to check in practice. Therefore, we also provide a sufficient, but not necessary, condition that is easier to verify and covers a large class of functions, e.g. all functions in Table 1. For functions in this class, we also provide an explicit construction of $u$, $\rho$, and $h_t$. See Appendix B for details.
> 3) When appending all past rewards to the state, the state space grows exponentially with the trajectory length, but the representation of a single state grows only linearly. Reinforcement learning methods have succesfully been applied to a variety of problems with inherently exponentially large state spaces, but modest trajectory lengths (e.g. Chess, Go). Therefore, appending all past rewards to the state may not be limiting in some problems (though it may be in others).
>
>
> **'The variables u, h and $\rho$ are discussed in the paragraph following the statements of equations 3,4,5. You should introduce them before [...] I suggest the statement of theorem 1 be rearranged to: assumptions then conclusion, instead of the current: assumption then conclusion then more assumptions.'**
>
> We prefer to first give the rigorous (and already self-contained) definition of these variables in Definition 1, and then provide intuition for this definition after that (note, that there are no additional assumptions after Definition 1). Finally, we present the result (Theorem 1) that builds on this definition. Ultimately, the choice of this structure is, of course, highly subjective.
> However, we agree that the discussion about constant size extra state information should not interrupt the argument between Definition 1 and Theorem 1 and moved it to the end of the theory section.
>
> **'[...]Also, their description is too vague. For example "This can be achieved by extending the state space with ht , which preserves all necessary information about the reward history" does not give me a good sense of what the function u should be.[...]'**
>
> The functions and variables are rigorously defined in Definition 1. The description in the paragraph after Definition 1 is simply to provide intuition. We agree that a more concrete statement would be desirable. However, this is unfortunately not possible for general $f$. Therefore, we moved the example with the minimum function right after this statement in the updated manuscript.  We also provide many examples in Table 1 (second row defines $u$, third row defines $\rho$). Additionally, we now provide a construction of $u$, $\rho$, and $h_t$ for a large class of objectives in Appendix B.

---

### Author Response · Authors · 2024-12-03

We are grateful to the reviewers for their constructive feedback and to the area, senior area, and program chairs for their time. **We believe that we have addressed all of the reviewers' major concerns:**
1) We show that the class of objectives $f$ that leads to constant size extra state information is **strictly more general than previous methods** (new Appendix B.1).
2) **We provide an explicit construction of the extra state information $h_t$** and the functions $u$ and $\rho$ for a large class of functions, e.g. all functions in Table 1 (new Appendix B.2).
3) We compare our method with the only previous method for solving NCMDPs with objectives other than the maximum function, which is only applicable in deterministic settings in conjunction with Q-learning based methods and a more restrictive class of functions 𝑓 than our method. We find that even in this restricted setting, **our more general method outperforms the previous method** (new Appendix C).
4) We show that **previous methods fail to solve the proof-of-principle example in Figure 2, while our method successfully finds the optimal policy** (Table A1, new Table A2).
5) We better motivate the choice of experiments, where we show that **in a variety of real-world tasks, performance can be improved over the previous state-of-the-art** by specifying the true non-cumulative objective to the agent (start of Section 3).

---

### Meta-Review · Area_Chair_evcw · 2024-12-23

**Metareview:**

The paper studies non cumulative Markov decision process (NCMDPs) in which the goal is to maximize the expected value of an arbitrary function of the rewards (not necessarily the sum of the rewards as in regular MDPs). The authors propose a method to map NCMDPs to regular MDPs and then use MDP solvers to solve them. They report experimental results in classical control, portfolio optimization, and discrete optimization with NCMDP objectives, and shows that their method improves both training time and final performance compared to using standard MDP solvers with cumulative reward.

The reviewers saw the followings as positives for the paper:
(+) The paper is well-written.
(+) The reviewers found the mapping from NCMDPs to standard MDPs simple and intuitive.

However, despite its simplicity, the method is not very practical because of
(-) The exponential growth of the state space with the trajectory length. This hugely restricts the applicability of the method.
The approach is not practical beyond the examples listed in the paper.

Moreover,
(-) The empirical comparisons with standard MDP solvers with cumulative reward do not make much sense and do not add much value.

**Additional Comments On Reviewer Discussion:**

The authors addressed a number of reviewers' concerns and some reviewers raised their scores. However, others still found the work slightly below the bar, due to the reasons listed in the meta-review.

---

> ### Public Comment · ~Maximilian_Nägele1 · 2025-02-10
>
> We thank the area chair for their time. However, we believe that there is a fundamental misunderstanding when it comes to the supposed drawbacks of our method:
>
> >'However, despite its simplicity, the method is not very practical because of (-) The exponential growth of the state space with the trajectory length. This hugely restricts the applicability of the method.'
>
> Exponential state spaces are not a 'huge restriction'. In fact, **exponentially growing (e.g. chess, go) or even infinite (e.g. continuous) state spaces are the norm in modern deep reinforcement learning**.
>
> >'The empirical comparisons with standard MDP solvers with cumulative reward do not make much sense and do not add much value.'
>
> The real-world applications we investigate in the empirical section involve non-cumulative objectives and stochastic environments that place them beyond the scope of previous NCMDP-solving methods. As a result, **previous state-of-the-art reinforcement learning approaches have been constrained by the need to approximate non-cumulative objectives** using standard MDPs. In contrast, **our method allows the exact objective to be specified to the reinforcement learning agent, resulting in improved performance**.

---

### Decision · Program_Chairs · 2025-01-22

Reject